# ADAPTIVE SELF-TRAINING FRAMEWORK FOR FINE-GRAINED SCENE GRAPH GENERATION

**Kibum Kim**[1]* **Kanghoon Yoon**[1]* **Yeonjun In**[1] **Jinyoung Moon**[2] **Donghyun Kim**[3]
**Chanyoung Park**[1]†
[1]KAIST  [2]ETRI  [3]Korea University
{kb.kim,ykhoon08,yeonjun.in,cy.park}@kaist.ac.kr
jymoon@etri.re.kr, d_kim@korea.ac.kr

## ABSTRACT

Scene graph generation (SGG) models have suffered from inherent problems regarding the benchmark datasets such as the long-tailed predicate distribution and missing annotation problems. In this work, we aim to alleviate the long-tailed problem of SGG by utilizing unannotated triplets. To this end, we introduce a *Self-Training framework for SGG* (ST-SGG) that assigns pseudo-labels for unannotated triplets based on which the SGG models are trained. While there has been significant progress in self-training for image recognition, designing a self-training framework for the SGG task is more challenging due to its inherent nature such as the semantic ambiguity and the long-tailed distribution of predicate classes. Hence, we propose a novel pseudo-labeling technique for SGG, called *Class-specific Adaptive Thresholding with Momentum* (CATM), which is a model-agnostic framework that can be applied to any existing SGG models. Furthermore, we devise a graph structure learner (GSL) that is beneficial when adopting our proposed self-training framework to the state-of-the-art message-passing neural network (MPNN)-based SGG models. Our extensive experiments verify the effectiveness of ST-SGG on various SGG models, particularly in enhancing the performance on fine-grained predicate classes. Our code is available on https://github.com/rlqja1107/torch-ST-SGG

## 1 INTRODUCTION

Scene graph generation (SGG) is a task designed to provide a structured understanding of a scene by transforming a scene of an image into a compositional representation that consists of multiple triplets in the form of ⟨subject, predicate, object⟩. Existing SGG methods have faced challenges due to inherent issues in the benchmark datasets (Krishna et al., 2017; Lu et al., 2016), such as the **long-tailed predicate distribution** and **missing annotations** for predicates (see Fig. 1). Specifically, the long-tailed predicate distribution in SGG refers to a distribution in which general predicates (e.g. "on") frequently appear, while fine-grained predicates (e.g. "walking in") are rarely present. Owing to such a long-tailed predicate distribution inherent in the dataset, existing SGG models tend to make accurate predictions for general predicates, while making incorrect predictions for fine-grained predicates. However, scene graphs primarily composed of general predicates are less informative in depicting a scene, which in turn diminishes their utility across a range of SGG downstream applications. Besides the challenge posed by the long-tailed predicate distribution, benchmark scene graph datasets (e.g., Visual Genome (VG) (Krishna et al., 2017)) encounter the problem of missing annotations. Specifically, missing annotations provide incorrect supervision (Zhang et al., 2020) to SGG models as unannotated triplets are carelessly assigned to the background class (i.e., bg), even though some unannotated triplets should have been indeed annotated with another class (See Fig. 1(a)). For example, treating the missing annotation between person and sidewalk as bg may confuse SGG models that are already trained with a triplet ⟨person, walking in, sidewalk⟩. In addition, the prevalence of triplets with the bg predicates between person and sidewalk throughout the dataset would exacerbate the problem. Addressing the missing annotation problem is especially important as a large of volume of benchmark scene graph datasets includes the bg class (e.g., 95.5% of the triplets are annotated with the bg class in VG dataset.

---

*Equal Contribution
†Corresponding Author

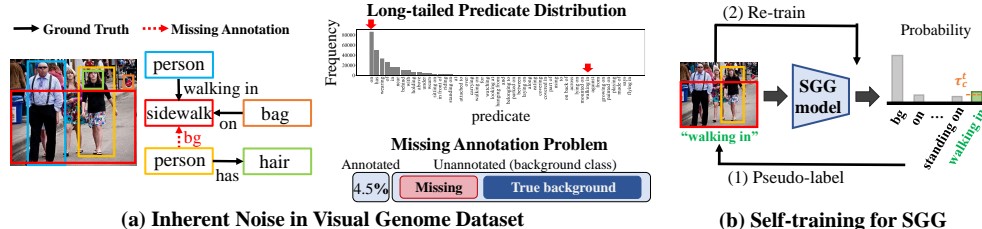

Figure 1: (a) Problems in VG scene graph dataset, and (b) self-training framework for SGG.

To alleviate the challenge posed by the long-tailed predicate distribution, recent SGG methods have presented resampling (Desai et al., 2021; Li et al., 2021), reweighting (Yan et al., 2020; Lyu et al., 2022), and various debiasing strategies (Tang et al., 2020; Guo et al., 2021; Chiou et al., 2021; Dong et al., 2022). However, these approaches simply adjust the proportion of the training data or the weight of the training objective without explicitly increasing the diversity of the data, which eventually leads to the model's overfitting to minority classes. IE-Trans (Zhang et al., 2022) is the most recent method proposed to increase the variation of the training data, either by replacing general predicates (i.e., the majority) with the fine-grained predicates (i.e., the minority) or by filling in missing annotations with fine-grained predicates. This study demonstrates that discovering informative triplets that have not been annotated helps alleviate the problem caused by the long-tailed predicate distribution, and improves the model's generalization performance. However, IE-trans involves many incorrect pseudo-labels in the training process as it relies solely on the initial parameters of the pre-trained model to generate pseudo-labels of the entire dataset in a one-shot manner. Hence, IE-trans fails to fully exploit the true labels of unannotated triplets.

In this paper, we aim to effectively utilize the unannotated triplets in benchmark scene graph datasets by assigning them accurate pseudo-labels. To this end, we introduce a self-training framework for SGG, called ST-SGG, which assigns pseudo-labels to confident predictions among unannotated triplets, and iteratively trains the SGG model based on them (Fig. 1.(b)). ST-SGG reduces the number of incorrect pseudo-labels by iteratively updating both the pseudo-labels and the SGG model at every batch step. This in turn mutually enhances the quality of pseudo-labels and the SGG model, which results in an effective use of unannotated triplets. While there has been significant progress in self-training for image recognition (Sohn et al., 2020; Xie et al., 2020b; Lee et al., 2013; Xu et al., 2021), designing a self-training framework for the SGG task is more challenging due to the uniqueness of the SGG nature, where the following factors that need to be considered when setting a proper threshold that determines confident predictions for unannotated triplets:

- **Semantic Ambiguity of Predicate Classes:** A group of predicate classes in benchmark scene graph datasets are highly related, e.g., "on", "standing on" and "walking on," which share similar semantics. This implies that the prediction probabilities for predicate classes are not sharpened for a specific predicate class (Fig. 1.(b)), and thus setting a proper threshold for determining a confident prediction is non-trivial. This problem becomes severe due to the fact that unannotated triplets in benchmark SGG datasets are annotated with the bg class, and this accounts for 95.5% of the entire triplets in VG dataset, resulting in SGG models to inevitably produce highly confident predictions for the bg class. Hence, pseudo-labeling methods used for image recognition (e.g., FixMatch (Sohn et al., 2020)) that assign the same threshold (e.g., 0.95) for all classes is not suitable for the SGG model (as demonstrated by Motif-$\tau^{\text{con}}$ in Sec. 3.3). Therefore, it is crucial to develop a pseudo-labeling strategy specifically designed for the SGG task, which is capable of autonomously determining the criteria for how reliable its model predictions are for each class, i.e., *class-specific adaptive threshold*.

- **Long-tailed Predicate Distribution:** The long-tailed nature of benchmark scene graph datasets poses a challenge in designing a self-training framework for SGG models. More precisely, an SGG model trained on long-tailed datasets tends to mainly generate highly confident pseudo-labels from the majority classes, and retraining the model with them would further exacerbate the bias towards the majority classes, causing self-training to fail (as demonstrated by Fig. 2 of Sec. 3.3). Therefore, for the SGG task, it is crucial to develop a pseudo-labeling strategy for self-training that assigns accurate pseudo-labels to the minority classes, while preventing bias towards the majority classes.

To cope with the above challenges of pseudo-labeling in SGG, we propose a novel thresholding strategy for the SGG task, called class-specific adaptive thresholding with momentum (CATM),

which adaptively adjusts the threshold by considering not only the bias towards majority classes, but also the highly related predicates and the presence of the bg class. In Sec. 3.3, we empirically demonstrate that the above factors are indeed mandatory for designing a self-training framework for SGG. It is worth noting that ST-SGG is a model-agnostic framework that can be applied to any existing SGG models including resampling and reweighting methods such as (Li et al., 2021) and (Sotiris Kotsiantis, 2006).

Additionally, we particularly focus on building upon message-passing neural networks (MPNN)-based SGG models, as they have recently shown to be the state-of-the-art (Li et al., 2021; Yoon et al., 2023). MPNN-based SGG models aim to learn structural relationships among entities based on the message propagation on a scene graph, which is shown to improve the representation quality of entities and relationships. We discovered that enriching the given scene graph structure, i.e., adding missing edges and removing noisy edges, is beneficial when adopting our proposed self-training framework to MPNN-based SGG models, particularly when setting the class-specific thresholds for pseudo-labeling unannotated triplets. Hence, we devise a graph structure learner (GSL) that learns to enrich the given scene graph structure, and incorporate it into our MPNN-based ST-SGG framework, which allows the messages to be propagated based on the enriched structure learned by the GSL. We show that this results in decreasing the model confidence on the bg class while increasing that on the other classes, both of which are helpful for setting the class-specific threshold. Through extensive experiments on VG and Open Images V6 (OI-V6), we verify that ST-SGG is effective when applied to existing SGG models, particularly enhancing the performance on fine-grained predicate classes.

- To the best of our knowledge, this is the first work that adopts self-training for SGG, which is challenging due to the difficulty in setting the threshold for pseudo-labeling in the SGG nature.

- We develop a novel thresholding technique CATM and the graph structure learner (GSL), which are beneficial for setting thresholds when ST-SGG is applied to any SGG models, e.g., Motif (Zellers et al., 2018) and VCTree (Tang et al., 2019), and MPNN-based models, e.g., BGNN (Li et al., 2021) and HetSGG (Yoon et al., 2023), respectively.

- Through extensive experiments on VG and OI-V6, we verify the effectiveness of ST-SGG compared with the state-of-the-art debiasing methods, particularly enhancing the performance on fine-grained predicate classes.

## 2 RELATED WORK

In this section, we briefly describe the related works and present the focus of our research. Please see Appendix. A for the extended explanation of related works.

**Self-Training.** In the field of image recognition, self-training is one of the prominent semi-supervised approaches that utilize a large amount of unlabeled samples. The main idea is to assign pseudo-labels to unlabeled samples whose confidence is above a specified threshold, and use them for model training to improve the generalization performance of the model. Recent works have focused on reliving the confirmation bias (Arazo et al., 2020) arising from using incorrect pseudo-labels for model training, which in turn undesirably increases the confidence of incorrect predictions. Hence, numerous confidence-based thresholding techniques are proposed to accurately assign pseudo-labels to unlabeled samples (Xie et al., 2020a; Sohn et al., 2020; Xu et al., 2021; Zhang et al., 2021; Lai et al., 2022; Guo et al., 2021; Wei et al., 2021). However, none of the above methods addresses the unique challenges of applying a self-training framework in the context of SGG (we will discuss more details of the challenges in Sec. 3.3). In this work, we propose a novel thresholding technique that considers the inherent nature of SGG, where the semantic ambiguity of predicates and the bias caused by long-tailed distribution are present.

**Scene Graph Generation.** Existing SGG methods point out that SGG models accurately predict general predicates, while rarely making correct predictions for fine-grained predicates. Desai et al. (2021); Li et al. (2021) and Desai et al. (2021); Li et al. (2021) aim to alleviate the long-tailed predicate distribution problem by using resampling and reweighting, respectively. Several debiasing methods are presented by Tang et al. (2020); Dong et al. (2022); Guo et al. (2021); Chiou et al. (2021). Most recently, IE-trans (Zhang et al., 2022) proposed to replace general predicates with fine-grained predicates and fill in missing annotations with fine-grained predicates to mitigate the problem caused by the long-tailed predicate distribution. However, IE-trans fails to fully exploit the true

labels of unannotated triplets as it uses the initial model parameter of the pre-trained model to generate pseudo-labels. On the other hand, we propose a novel pseudo-labeling technique for SGG to effectively utilize the unannotated triplets, which helps reduce the number of incorrect pseudo-labels by iteratively updating both the pseudo-labels and the SGG model. Further, we extend ST-SGG for message-passing neural network (MPNN)-based SGG models (Zellers et al., 2018; Tang et al., 2019; Xu et al., 2017; Yang et al., 2018; Lin et al., 2020; Li et al., 2021; Yoon et al., 2023; Shit et al., 2022), which are considered as state-of-the-art SGG models utilizing the advanced SGG architecture.

# 3 SELF-TRAINING FRAMEWORK FOR SGG (ST-SGG)

## 3.1 PRELIMINARIES

**Notations.** Given an image $\mathbf{I}$, a scene graph $\mathbf{G}$ is represented as a set of triplets $\{(\mathbf{s}_i, \mathbf{p}_i, \mathbf{o}_i)\}_{i=1}^{M}$, where $M$ is the number of triplets in $\mathbf{G}$. A subject $\mathbf{s}_i$ is associated with a class label $\mathbf{s}_{i,c} \in \mathcal{C}^e$ and a bounding box position $\mathbf{s}_{i,b} \in \mathbb{R}^4$, where $\mathcal{C}^e$ is the set of possible classes for an entity. Likewise, an object $\mathbf{o}_i$ is associated with $\mathbf{o}_{i,c} \in \mathcal{C}^e$ and $\mathbf{o}_{i,b} \in \mathbb{R}^4$. A predicate $\mathbf{p}_i$ denotes the relationship between $\mathbf{s}_i$ and $\mathbf{o}_i$, and it is associated with a class label $\mathbf{p}_{i,c} \in \mathcal{C}^p$, where $\mathcal{C}^p$ is the set of possible classes for predicates. Note that $\mathcal{C}^p$ includes the "background" class, which represents there exists no relationship between $\mathbf{s}_i$ and $\mathbf{o}_i$ (i.e., $\mathbf{p}_{i,c} = \mathsf{bg}$).

**SGG Task.** Our goal is to train an SGG model $f_\theta : \mathbf{I} \to \mathbf{G}$, which generates a scene graph $\mathbf{G}$ from an image $\mathbf{I}$. Generally, an SGG model first generates entity and relation proposals using a pre-trained object detector such as Faster R-CNN (Ren et al., 2015). Specifically, an entity proposal $\mathbf{x}^e$ is represented by the output of feedforward network that takes a bounding box, its visual feature, and the word embedding of the entity class. The predicted probability of the entity class $\hat{\mathbf{p}}^e \in \mathbb{R}^{|\mathcal{C}^e|}$ is estimated by an entity classifier $f_\theta^e$ (i.e., $\hat{\mathbf{p}}^e = f_\theta^e(\mathbf{x}^e)$). Moreover, a relation proposal $\mathbf{x}^{\mathbf{s},\mathbf{o}}$ between two entities (i.e., subject $\mathbf{s}$ and object $\mathbf{o}$) is represented by the union box of the visual feature of the two entity proposals. The predicted probability of the predicate class $\hat{\mathbf{p}}^p \in \mathbb{R}^{|\mathcal{C}^p|}$ between two entities is estimated by a predicate classifier $f_\theta^p$ (i.e., $\hat{\mathbf{p}}^p = f_\theta^p(\mathbf{x}^{\mathbf{s},\mathbf{o}})$). Finally, the SGG model selects the most probable triplets $\{(\mathbf{s}_i, \mathbf{p}_i, \mathbf{o}_i)\}_{i=1}^{M}$ as a generated scene graph.

## 3.2 PROBLEM FORMULATION OF ST-SGG

We introduce a self-training framework for SGG (ST-SGG), aiming to exploit a large volume of unannotated triplets, which is formulated under the setting of semi-supervised learning.

Given $B$ batches of scene graphs $\{\mathbf{G}_1, ..., \mathbf{G}_b, ..., \mathbf{G}_B\}$, each batch $\mathbf{G}_b$ contains a set of annotated triplets $\mathbf{G}_b^A = \{(\mathbf{s}_i, \mathbf{p}_i, \mathbf{o}_i)|\mathbf{p}_i \neq \mathsf{bg}\}$, and a set of unannotated triplets $\mathbf{G}_b^U = \{(\mathbf{s}_i, \mathbf{p}_i, \mathbf{o}_i)|\mathbf{p}_i = \mathsf{bg}\}$. Note that only a small portion of triplets is annotated, i.e., $|\mathbf{G}_b^U| \gg |\mathbf{G}_b^A|$. For example, only 4.5% of relationships between objects are annotated in the VG dataset (Krishna et al., 2017).

To exploit the unannotated triplets, we adopt a self-training approach for SGG that assigns pseudo-labels to predicates in the unannotated triplets. Given a model prediction $\hat{\mathbf{p}}^p \in \mathbb{R}^{|\mathcal{C}^p|}$, we define a *confidence* $\hat{q}$ of the model prediction by the maximum value of $\hat{\mathbf{p}}^p$, i.e., $\hat{q} = \max(\hat{\mathbf{p}}^p) \in \mathbb{R}$, and denote the corresponding predicate class as $\tilde{\mathbf{q}} \in \mathbb{R}^{|\mathcal{C}^p|}$, which is a one-hot vector. We assign $\tilde{\mathbf{q}}$ to predicates in the set of unannotated triplets $\mathbf{G}_b^U$ if the model confidence is greater than a threshold $\tau$, and otherwise leave them as $\mathsf{bg}$. Given the labeled triplets in $\mathbf{G}_b^A$ and pseudo-labeled triplets in $\mathbf{G}_b^U$, we retrain the SGG model at every batch step (i.e., iteration). Formally, the loss for training the SGG model under the self-training framework is divided into the following three losses:

$$\mathcal{L} = \mathbb{E}_{1 \leq b \leq B}\left[\underbrace{\mathbb{E}_{i \in \mathbf{G}_b^A}\left[-\mathbf{p}_i \cdot \log(\hat{\mathbf{p}}_i^p)\right]}_{\text{Loss for annotated predicates}} + \underbrace{\mathbb{E}_{i \in \mathbf{G}_b^U | \hat{q}_i < \tau}\left[-\mathbf{p}_i^{\mathsf{bg}} \cdot \log(\hat{\mathbf{p}}_i^p)\right]}_{\text{Loss for } \mathsf{bg} \text{ class}} + \beta\underbrace{\mathbb{E}_{i \in \mathbf{G}_b^U | \hat{q}_i \geq \tau}\left[-\tilde{\mathbf{q}}_i \cdot \log(\hat{\mathbf{p}}_i^p)\right]}_{\text{Loss for pseudo-labeled predicates}}\right],$$

(1)

where $\beta$ is a coefficient for controlling the weight of the pseudo-label loss, and $\mathbf{p}^{\mathsf{bg}} \in \mathbb{R}^{|\mathcal{C}^p|}$ is a one-hot vector that represents the $\mathsf{bg}$ class. Despite the fact that there may exist true relationships rather than $\mathsf{bg}$ in the unannotated triplets, most existing SGG models simply consider all the unannotated triplets as belonging to the $\mathsf{bg}$ class, which means the third loss is merged with the second loss (with $\beta = 1$) in Equation 1. It is important to note that ST-SGG is a model-agnostic framework that can be applied to any existing SGG models.

## 3.3 CHALLENGES OF SELF-TRAINING FOR SGG

Training an SGG model based on Equation 1 facilitates the self-training of the model by exploiting unannotated triplets with pseudo-labels. However, unless the threshold $\tau$ is carefully set, SGG models deteriorate since they are prone to confirmation bias, which is caused by confidently assigning incorrect pseudo-labels to unannotated triplets. In Fig. 2, we empirically investigate the effectiveness of various thresholding techniques by applying them to ST-SGG. Specifically, we first pre-train Motif (Zellers et al., 2018) on the VG dataset, and subsequently fine-tune it using Equation 1 while employing various thresholding techniques including constant thresholding (Sohn et al., 2020) (i.e., $\tau = \tau^{\mathrm{con}}$), fixed class-specific thresholding (i.e., $\tau = \tau_c^{\mathrm{cls}}$), class frequency-weighted fixed class-specific thresholding (Wei et al., 2021) that considers the long-tailed distribution (i.e., $\tau = \tau_c^{\mathrm{lt}}$), and class-specific adaptive thresholding (Xu et al., 2021) (i.e., $\tau = \tau_c^{\mathrm{ada}}$). Please refer to Appendix B.2 for details on the thresholding techniques.

We have the following three observations: **1)** Motif-$\tau^{\mathrm{con}}$, Motif-$\tau_c^{\mathrm{cls}}$ and Motif-$\tau_c^{\mathrm{lt}}$ show lower performance than even Motif-vanilla (without self-training) over all metrics, failing to benefit from the unannotated triplets through self-training. This implies that thresholding techniques that are widely

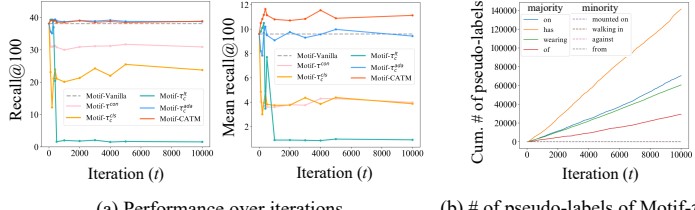

(a) Performance over iterations          (b) # of pseudo-labels of Motif-$\tau_c^{\mathrm{cls}}$

Figure 2: (a) Performance of Motif and its self-trained models with various thresholding techniques. (b) The number of pseudo-labels per predicate class when Motif-$\tau_c^{\mathrm{cls}}$ is trained on VG.

used in image recognition are not suitable for ST-SGG. We argue that this is mainly due to highly related predicate classes, the presence of bg class, and the long-tailed distribution. As these problems co-exist, determining an appropriate threshold becomes difficult when adopting self-training to SGG models. **2)** In Fig. 2(b), we further report the behavior of Motif-$\tau_c^{\mathrm{cls}}$ to investigate the reason for its failure. We observe that as the training progresses, the number of majority predicate classes being assigned as pseudo-labels increases, while minority predicate classes are not assigned at all. This implies that although we set the different thresholds for the majority and minority predicate classes to address the long-tailed problem, it is non-trivial due to the fact that other issues, such as the semantic ambiguity of predicate classes and the presence of the bg classs, are entangled with the setting of a proper threshold in the nature of SGG. We observe similar behaviors for Motif-$\tau^{\mathrm{con}}$ and Motif-$\tau_c^{\mathrm{lt}}$. **3)** The performance of Motif-$\tau_c^{\mathrm{ada}}$ decreases at the early stage of training, but its final performance is similar to that of Motif-Vanilla at the later stage. We attribute this to the incorrect pseudo-label assignment in the early training steps and the saturation of the adaptive threshold in the later training steps, which prevents the model from further assigning any pseudo-labels. This implies that a naive implementation of the class-specific adaptive thresholding technique fails to fully exploit the unannotated triplets in self-training.

In summary, it is challenging to determine an appropriate threshold for adopting ST-SGG due to the inherent nature of SGG, mainly incurred by the semantic ambiguity of predicates and the long-tailed predicate distribution.

## 4 CLASS-SPECIFIC ADAPTIVE THRESHOLDING WITH MOMENTUM (CATM)

To address the aforementioned challenges of applying self-training to the SGG task, we devise a novel pseudo-labeling technique for SGG, called *Class-specific Adaptive Thresholding with Momentum* (CATM), which adaptively adjusts the class-specific threshold by considering not only the long-tailed predicate distribution but also the model's learning state of each predicate class.

### 4.1 CLASS-SPECIFIC ADAPTIVE THRESHOLDING

Our approach to adjusting the threshold involves using the model's confidence of the predicate prediction on unannotated triplets, as these predictions reflect the per class learning state of the model. We argue that relying on the model prediction to determine the threshold has two advantages. First, it addresses the long-tailed nature of predicates that causes significant differences in learning states across predicate classes. Second, although model prediction probabilities for a certain predicate

may not be sharpened (or confident) due to the semantic ambiguity of predicate classes, it enables us to set an appropriate threshold value based on the model predictions on other instances of the same predicate class. A straightforward approach to estimating the model prediction-based threshold would be to compute the average of the model's confidence in the validation set, and set it as the threshold at every iteration or at a regular iteration interval. However, it requires a significant computational cost, and the learning state of the model between iteration intervals can not be reflected (Refer to Appendix C.2).

To this end, we employ the exponential moving average (EMA) (Wang et al., 2022) to adjust the threshold at each iteration, in which the threshold is either increased or decreased depending on the confidence of the prediction and the threshold at the previous iteration. More formally, we use $\tau_c^t$ to denote the threshold of a predicate class $c$ at iteration $t$, and use $\mathcal{P}_c^U$ to denote the set of predicates in unannotated triplets in all $B$ batches, i.e., $\{\mathbf{G}_1^U, ..., \mathbf{G}_b^U, ..., \mathbf{G}_B^U\}$, that are predicted as belonging to the predicate class $c$. Our main goal is to assign pseudo-labels to unannotated triplets while filtering out incorrect pseudo-labels. Specifically, we increase the threshold $\tau_c^t$ for a given a predicate class $c$, if the confidence $\hat{q}$ of the prediction is greater than the previous threshold $\tau_c^{t-1}$ (See the first condition in Equation 2). On the other hand, if the confidence $\hat{q}$ for all instances of the predicate class $c$ is lower than the previous threshold $\tau_c^{t-1}$, we decrease the threshold for the predicate class $c$, since the current threshold $\tau_c^t$ is likely to be over-estimated (See the second condition in Equation 2). The threshold of the predicate class $c$ at iteration $t$, i.e., $\tau_c^t$, is updated based on EMA as follows:

$$\tau_c^t = \begin{cases} (1 - \lambda^{inc}) \cdot \tau_c^{t-1} + \lambda^{inc} \cdot \mathbb{E}_{i \in \mathcal{P}_c^U} [\hat{q}_i], & \text{if } \exists i \in \mathcal{P}_c^U \text{ where } \hat{q}_i \geq \tau_c^{t-1}. (\tau_c^{t-1} \text{ increases}) \\ (1 - \lambda^{dec}) \cdot \tau_c^{t-1} + \lambda^{dec} \cdot \mathbb{E}_{i \in \mathcal{P}_c^U} [\hat{q}_i], & \text{if } \hat{q}_i < \tau_c^{t-1} \text{ for all } i \in \mathcal{P}_c^U. (\tau_c^{t-1} \text{ decreases}) \\ \tau_c^{t-1}, & \text{if } \mathcal{P}_c^U = \emptyset. (\tau_c^{t-1} \text{ remains}) \end{cases} \quad (2)$$

where $\lambda^{inc}$ and $\lambda^{dec}$ are the momentum coefficients for increasing and decreasing the threshold, respectively. Note that $\tau_c^t$ is updated only if there exists at least one predicate that is predicted as belonging to the predicate class $c$ in any of the batches, and otherwise it is maintained (See the third condition in Equation 2).

As the class-specific threshold reflects the learning state of the model for each predicate class at each iteration, the above EMA approach that adaptively adjusts the class-specific threshold captures the differences in the learning states of the model across the predicate classes. This in turn allows the self-training framework to be applied in the SGG nature. Moreover, the EMA approach is superior to the naive method as it establishes a more stable threshold by considering the model's learning state at every iteration and the accumulated confidence in samples. It is also computationally efficient and does not require computing the confidence on additional datasets, such as a validation set.

## 4.2 CLASS-SPECIFIC MOMENTUM

However, we observed that if the same momentum hyperparameter is used for all predicate classes (i.e., setting the same $\lambda^{inc}$ and $\lambda^{dec}$ for all $c \in \mathcal{C}^p$), unannotated triplets are mainly pseudo-labeled with majority predicate classes, whereas minority predicate classes receive less attention (See Fig. 9(a) in Appendix E.7). We argue that this incurs confirmation bias and aggravates the problem of long-tailed predicate distribution in SGG. To cope with this challenge, we design the *class-specific momentum*, which sets different $\lambda_c^{inc}$ and $\lambda_c^{dec}$ for each $c \in \mathcal{C}^p$ based on the frequency of predicate classes in the training set. The main idea is to increase the threshold of majority predicate classes more rapidly than that of minority predicate classes so that unannotated triplets are pseudo-labeled with minority predicate classes as the training progresses. Conversely, when decreasing the threshold, we want the threshold of majority predicate classes to decrease more slowly than that of minority predicate classes. More formally, let $N_1, ..., N_{|\mathcal{C}^p|}$ be the number of instances that belong to the predicate classes $c_1, ...c_{|\mathcal{C}^p|}$ in the training set, respectively, which is sorted in descending order (i.e., $N_1 > N_2 > ... > N_{|\mathcal{C}^p|}$). We set the increasing momentum coefficient and decreasing momentum coefficient as $\lambda_c^{inc} = \left(\frac{N_c}{N_1}\right)^{\alpha^{inc}}$, $\lambda_c^{dec} = \left(\frac{N_{(|\mathcal{C}^p|+1-c)}}{N_1}\right)^{\alpha^{dec}}$, where $\alpha^{inc}, \alpha^{dec} \in [0, 1]$ control the increasing and decreasing rate, respectively, and $\frac{N_c}{N_1}$ and $\frac{N_{(|\mathcal{C}^p|+1-c)}}{N_1}$ are the imbalance ratio and reverse imbalance ratio of class $c$. For example, assume that we are given three classes with $[N_1, N_2, N_3] = [50, 40, 10]$ where class 1 and 2 are head predicate classes, and class 3 is a tail predicate class. In this case, the imbalance ratio is $[1.0, 0.8, 0.2]$, and the reverse imbalance ratio is $[0.2, 0.8, 1.0]$. Hence, the increasing rates of class 1 and class 3 are

$\lambda_1^{inc} = (1.0)^{\alpha^{inc}}$ and $\lambda_3^{inc} = (0.2)^{\alpha^{inc}}$, respectively, which implies that the threshold of class 1 increases more rapidly than that of class 3. On the other hand, the decreasing rates of class 1 and class 3 are $\lambda_1^{dec} = (0.2)^{\alpha^{dec}}$ and $\lambda_3^{inc} = (1.0)^{\alpha^{dec}}$, respectively, which implies that the threshold of class 1 decreases more slowly than that of class 3. Please refer to Fig. 10 of Appendix E.8 regarding the change of $\lambda_c^{inc}$ and $\lambda_c^{dec}$ based on $\alpha^{inc}$ and $\alpha^{dec}$. As a result of setting the class-specific momentum coefficients, a small number of pseudo-labels are being assigned to head predicate classes, while tail predicate classes are assigned more actively (See Fig. 9(b) of Appendix E.7), which relieves the long-tailed the problem of predicates. We believe that pseudo-labeling the unannotated triplets with more tail predicate classes than head predicate classes is acceptable since the original training set already contains a large volume of instances that belong to head predicate classes.

### 4.3 GRAPH STRUCTURE LEARNER FOR CONFIDENT PSEUDO-LABELS

We discovered that enriching the given scene graph structure, i.e., adding missing edges and removing noisy edges, is beneficial when adopting our self-training framework to MPNN-based SGG models. In Fig. 3, we report the confidence of predicate classes after training BGNN (Li et al., 2021), a SOTA MPNN-based SGG method, on the fully connected scene graph (i.e., BGNN w/o GSL) and on a scene graph whose structure

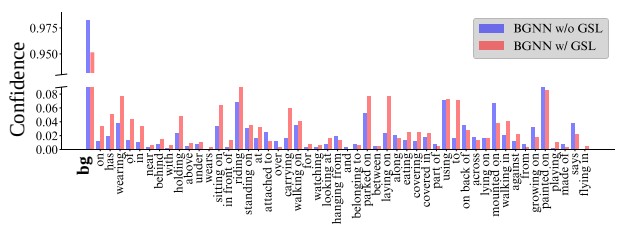

Figure 3: Impact of applying GSL on the model confidence of BGNN (Li et al., 2021).

is enriched by a graph structure learner (i.e., BGNN w/ GSL). A graph structure learner (GSL) is a method to learn relevant and irrelevant relations between entities in a scene graph, allowing MPNN-based methods to propagate messages only through relevant relations. We observe that BGNN with GSL generates relatively low confident predictions on the "bg" class, allowing the model to make more confident predictions on the remaining predicate classes, which is particularly beneficial when setting the class-specific thresholds for pseudo-labeling unannotated triplets. To this end, we devise a GSL that learns to enrich the given scene graph structure, and incorporate it into our MPNN-based ST-SGG framework. Please refer to Appendix D for more details. Note that we consider *only the relevant relations predicted by GSL as candidates for pseudo-labeling.*

## 5 EXPERIMENT

We compare ST-SGG with state-of-the-arts methods that alleviate the long-tailed problem on commonly used benchmark datasets, VG and OI-V6. We report only the result on VG due to the page limit. Please refer to the result on OI-V6 in Appendix F.3. More details of the experimental setups are described in Appendix E and F. Note that ST-SGG involves CATM by default.

### 5.1 COMPARISON WITH BASELINES ON VISUAL GENOME

In Table 1, we apply ST-SGG to widely-used SGG models, such as Motif (Zellers et al., 2018) and VCTree (Tang et al., 2019), and compare them with baselines. Based on the result, we have the following observations: **1)** ST-SGG exhibits model-agnostic adaptability to SGG models. Motif+ST-SGG and VCTree+ST-SGG improve their performance in terms of mR@K and F@K, implying that ST-SGG greatly increases the performance on tail predicates while retaining that of head predicates. **2)** ST-SGG that employs debiasing methods is competitive with the-state-of-art SGG models. Specifically, we employ resampling and I-Trans (Zhang et al., 2022) to Motif under ST-SGG. In terms of F@K, Motif+Resamp.+ST-SGG and Motif+I-Trans+ST-SGG outperform the performance of DT2-ACBS and PCPL in SGCls and SGDet despite the simple architecture of Motif. This implies that utilizing unannotated triplets with simple debiasing methods is powerful without advancing the architecture of SGG models. **3)** Compared to a previous pseudo-labeling method, i.e., IE-Trans, ST-SGG+I-Trans achieves better performance in terms of mR@K and F@K. This demonstrates that ST-SGG assigns accurate pseudo-labels to unannotated triplets, and relieves the long-tailed problem. We attribute this to the fact that pseudo-labeling in an iterative manner is more accurate than pseudo-labeling in a one-shot manner.

Table 1: Performance (%) comparison of ST-SGG with state-of-the art SGG models on three tasks. *Resam.* denotes the re-sampling (Li et al., 2021) method. The best performance is shown in bold.

| | Method | PredCls | | | SGCls | | | SGDet | | |
|---|---|---|---|---|---|---|---|---|---|---|
| | | R@50/100 | mR@50/100 | F@50/100 | R@50/100 | mR@50/100 | F@50/100 | R@50/100 | mR@50/100 | F@50/100 |
| Specific | DT2-ACBS (Desai et al., 2021) | 23.3/25.6 | 35.9/39.7 | 28.3/31.1 | 16.2/17.6 | 24.8/27.5 | 19.6/21.5 | 15.0/16.3 | 22.0/24.0 | 17.8/19.4 |
| | PCPL (Yan et al., 2020) | 50.8/52.6 | 35.2/37.8 | 41.6/44.0 | 27.6/28.4 | 18.6/19.6 | 22.2/23.2 | 14.6/18.6 | 9.5/11.7 | 11.5/14.4 |
| | KERN (Chen et al., 2019) | 65.8/67.6 | 17.7/19.2 | 27.9/29.9 | 36.7/37.4 | 9.4/10.0 | 15.0/15.8 | 27.1/29.8 | 6.4/7.3 | 10.4/11.7 |
| | GBNet (Zareian et al., 2020) | 66.6/68.2 | 22.1/24.0 | 33.2/35.5 | 37.3/38.0 | 12.7/13.4 | 18.9/19.8 | 26.3/29.9 | 7.1/8.5 | 11.2/13.2 |
| | PE-Net (Zheng et al., 2023) | 64.9/67.2 | 31.5/33.8 | 42.4/45.0 | 39.4/40.7 | 17.8/18.9 | 24.5/25.8 | 30.7/35.2 | 12.4/14.5 | 17.7/20.5 |
| Model-Agnostic | Motif (Zellers et al., 2018) | **65.3/67.1** | 17.8/19.2 | 28.0/29.9 | **36.9/38.1** | 9.0/9.6 | 14.5/15.3 | **31.9/36.4** | 6.4/7.6 | 10.7/12.6 |
| | +ST-SGG | 63.4/65.4 | 22.4/24.1 | 33.1/35.2 | 36.8/37.8 | 12.1/12.8 | 18.2/19.1 | 29.7/34.8 | 8.5/10.1 | 13.2/15.7 |
| | +Resam. (Li et al., 2021) | 62.3/64.3 | 26.1/28.5 | 36.8/39.5 | 36.1/37.0 | 13.7/14.7 | 19.9/21.0 | 30.4/34.8 | 10.5/12.3 | 15.6/18.2 |
| | +Resam.+ST-SGG | 53.9/57.7 | 28.1/31.5 | 36.9/40.8 | 33.4/34.9 | 16.9/18.0 | 22.4/23.8 | 26.7/30.7 | 11.6/14.2 | 16.2/19.4 |
| | +TDE (Tang et al., 2020) | 46.2/51.4 | 25.5/29.1 | 32.9/37.2 | 27.7/29.9 | 13.1/14.9 | 17.8/19.9 | 16.9/20.3 | 8.2/9.8 | 11.0/13.2 |
| | +DLFE (Chiou et al., 2021) | 52.5/54.2 | 26.9/28.8 | 35.6/37.6 | 32.3/33.1 | 15.2/15.9 | 20.7/21.5 | 25.4/29.4 | 11.7/13.8 | 16.0/18.8 |
| | +NICE (Li et al., 2022) | 55.1/57.2 | 29.9/32.3 | 38.8/41.3 | 33.1/34.0 | 16.6/17.9 | 22.1/23.5 | 27.8/31.8 | 12.2/14.4 | 17.0/19.8 |
| | +IE-Trans (Zhang et al., 2022) | 54.7/56.7 | 30.9/33.6 | 39.5/42.2 | 32.5/33.4 | 16.8/17.9 | 22.2/23.3 | 26.4/30.6 | 12.4/14.9 | 16.9/20.0 |
| | +I-Trans (Zhang et al., 2022) | 55.2/57.1 | 29.1/31.9 | 38.1/40.9 | 32.5/33.4 | 15.7/16.9 | 21.2/22.4 | 27.0/31.3 | 11.4/14.0 | 16.0/19.3 |
| | +I-Trans+ST-SGG | 50.5/52.8 | **32.5/35.1** | **41.7/42.5** | 31.2/32.1 | **18.0/19.3** | **22.8/24.1** | 25.7/29.8 | **12.9/15.8** | **17.2/20.7** |
| | VCTree (Tang et al., 2019) | **65.5/67.2** | 17.2/18.6 | 27.3/29.1 | **38.1/38.8** | 9.6/10.2 | 15.3/16.2 | **31.4/35.7** | 7.3/8.6 | 11.9/13.9 |
| | +ST-SGG | 64.2/66.2 | 21.5/22.9 | 32.2/34.0 | 37.5/38.4 | 12.0/12.5 | 18.2/18.9 | 30.4/34.7 | 8.7/10.1 | 13.5/15.6 |
| | +Resam. (Li et al., 2021) | 61.2/63.5 | 27.2/29.2 | 37.7/40.0 | 35.7/36.5 | 13.8/14.4 | 19.9/20.7 | 29.9/33.9 | 10.2/11.8 | 15.2/17.5 |
| | +Resam.+ST-SGG | 54.0/57.0 | 32.2/34.6 | **40.3/43.0** | 32.2/33.4 | 16.9/18.3 | 22.2/23.6 | 24.6/29.6 | 12.3/14.8 | **16.4/19.7** |
| | +TDE (Tang et al., 2020) | 47.2/51.6 | 25.4/28.7 | 33.0/36.9 | 25.4/27.9 | 12.2/14.0 | 16.5/18.6 | 19.4/23.2 | 9.3/11.1 | 12.6/15.0 |
| | +DLFE (Chiou et al., 2021) | 51.8/53.5 | 25.3/27.1 | 34.0/36.0 | 33.5/34.6 | 18.9/20.0 | 24.2/25.3 | 22.7/26.3 | 11.8/13.8 | 15.5/18.1 |
| | +NICE (Li et al., 2022) | 55.0/56.9 | 30.7/33.0 | 39.4/41.8 | 37.8/**39.0** | 19.9/21.3 | 26.1/27.6 | 27.0/30.8 | 11.9/14.1 | 16.5/19.3 |
| | +IE-Trans (Zhang et al., 2022) | 53.0/55.0 | 30.3/33.9 | 38.6/41.9 | 32.9/33.8 | 16.5/18.1 | 22.0/23.6 | 25.4/29.3 | 11.5/14.0 | 15.8/18.9 |
| | +I-Trans (Zhang et al., 2022) | 54.0/55.9 | 30.2/33.1 | 38.7/41.6 | 37.2/38.3 | 19.0/20.6 | 25.1/26.8 | 25.5/29.4 | 11.2/13.7 | 15.6/18.7 |
| | +I-Trans+ST-SGG | 52.5/54.3 | **32.7/35.6** | **40.3/43.0** | 36.3/37.3 | **21.0/22.4** | **26.6/27.9** | 20.7/24.9 | **12.6/15.1** | 15.7/18.8 |

Table 2: Performance (%) comparison of ST-SGG with state-of-the-art MPNN-based SGG models on three tasks. *Resam.* denotes the re-sampling (Li et al., 2021) method.

| | Method | PredCls | | | SGCls | | | SGDet | | |
|---|---|---|---|---|---|---|---|---|---|---|
| | | R@50/100 | mR@50/100 | F@50/100 | R@50/100 | mR@50/100 | F@50/100 | R@50/100 | mR@50/100 | F@50/100 |
| MPNN-based Model | BGNN+Resam. (Li et al., 2021) | 57.8/60.0 | 29.2/31.7 | 38.8/41.5 | **36.9/38.1** | 14.6/16.0 | 20.9/22.5 | | 11.4/13.3 | 16.5/19.2 |
| | +ST-SGG | 48.0/51.4 | 33.0/35.1 | 39.1/41.7 | 32.3/34.0 | 17.9/19.0 | 23.1/24.4 | 20.6/27.3 | 13.6/16.1 | 16.4/20.3 |
| | +ST-SGG+GSL | 48.9/51.4 | **34.1/36.2** | **40.2/42.5** | 33.5/34.7 | 18.0/19.4 | **23.4/24.9** | 26.5/31.4 | **14.1/16.6** | **18.4/21.7** |
| | BGNN+IE-Trans (Zhang et al., 2022) | 54.5/56.6 | 29.7/32.4 | 38.4/41.2 | 33.2/34.1 | 16.3/17.6 | 21.9/23.2 | 24.2/27.9 | 11.2/13.7 | 15.3/18.4 |
| | BGNN+I-Trans (Zhang et al., 2022) | 54.9/57.0 | 28.7/31.6 | 37.7/40.7 | 33.4/34.5 | 15.5/16.9 | 21.2/22.7 | 24.2/28.1 | 10.5/13.2 | 14.6/18.0 |
| | BGNN+I-Trans+ST-SGG+GSL | 52.7/54.7 | 31.5/34.5 | 39.4/42.3 | 31.4/32.3 | 17.6/18.9 | 22.6/23.9 | 22.6/26.2 | 12.0/14.6 | 15.7/18.8 |
| | HetSGG+Resam. (Yoon et al., 2023) | **58.0/60.1** | 30.0/32.2 | 39.5/41.9 | **37.6/38.5** | 15.8/17.7 | 22.2/24.3 | **30.2/34.5** | 11.5/13.5 | 16.7/19.4 |
| | +ST-SGG | 49.5/52.5 | 32.6/35.2 | 39.3/42.1 | 33.2/34.7 | 18.0/18.9 | 23.4/24.5 | 20.2/27.2 | 11.9/14.6 | 15.0/19.0 |
| | +ST-SGG+GSL | 50.2/53.2 | **33.6/35.8** | **40.3/42.8** | 35.5/36.5 | 18.2/19.1 | **24.1/25.1** | | **12.9/15.0** | **17.4/20.4** |
| | HetSGG+IE-Trans (Zhang et al., 2022) | 53.1/55.1 | 31.3/34.5 | 39.4/42.4 | 33.9/34.9 | 16.7/18.1 | 22.4/23.8 | 23.8/27.4 | 11.5/14.0 | 15.5/18.5 |
| | HetSGG+I-Trans (Zhang et al., 2022) | 52.1/54.1 | 30.8/34.2 | 38.7/41.9 | 33.0/33.9 | 16.3/17.7 | 21.8/23.3 | 23.9/27.6 | 10.7/13.7 | 14.8/18.3 |
| | HetSGG+I-Trans+ST-SGG+GSL | 49.6/51.4 | 32.9/**35.9** | 39.6/42.3 | 31.2/32.2 | 18.0/**19.3** | 22.8/24.1 | 22.3/25.6 | 12.3/14.9 | 15.9/18.8 |

Moreover, we investigate the result for each predicate. Fig. 4 shows that ST-SGG greatly improves the performance for the tail predicates while retaining that for the head predicates by pseudo-labeling the unannotated triplet to a tail predicates. It is worth noting that predicting fine-grained predicates (i.e., tail predicates) is more

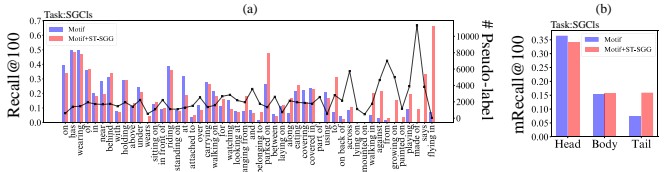

Figure 4: (a) Performance comparison per class. The black line indicates the number of pseudo-labeled instances. (b) Performance comparison on head, body, and tail predicate classes.

important than predicting general predicates as tail predicates are informative in depicting a scene. In this regard, ST-SGG is an effective framework that generates informative scene graphs by utilizing unannotated triplets.

## 5.2 COMPARISON WITH MPNN-BASED MODELS ON VISUAL GENOME

Table 2 shows the result of the state-of-art MPNN-based SGG models. Herein, we apply the graph structure learner (GSL) to existing MPNN-based models including BGNN (Li et al., 2021) and HetSGG (Yoon et al., 2023). We have the following observations: **1)** The mR@K and F@K of BGNN+ST-SGG and HetSGG+ST-SGG are increased compared to those of BGNN and HetSGG, respectively, which implies that ST-SGG effectively utilizes unannotated triplets when applied to MPNN-based SGG models. **2)** When MPNN-based models employ ST-SGG +GSL, they achieve state-of-art performance among SGG models, outperforming the models that solely use ST-SGG. This implies that GSL identifies relevant relationships, and the relations to which pseudo-labels are to be assigned.

## 5.3 ABLATION STUDY ON MODEL COMPONENTS OF ST-SGG

In Table 3, we ablate the component of ST-SGG to analyze the effect of each component. We select Motif and BGNN trained with resampling (Li et al., 2021) as the backbone SGG model, and train the following models for SGCls task. 1) ST-SGG w/o EMA: we remove the EMA of adaptive threshold, and apply the fixed threshold $\tau_c^{\text{cls}}$ used in Sec. 3.3. 2) ST-SGG w/o $\lambda_c^{inc}, \lambda_c^{dec}$: we remove

the class-specific momentum by setting $\lambda_c^{inc}$ and $\lambda_c^{dec}$ to 0.5. When the Motif backbone is used, we observed that the performance of ST-SGG w/o EMA severely decreases in terms of R@K and mR@K compared to Vanilla Motif since the fixed threshold cannot properly assign correct pseudo-labels. This implies that adaptively updating the threshold through EMA is important for ST-SGG. Beside, compared with ST-SGG, ST-SGG w/o $\lambda_c^{inc}, \lambda_c^{dec}$ shows

Table 3: Ablation study of ST-SGG.

| Back-bone | Model | SGCls | | |
|---|---|---|---|---|
| | | R@50 / 100 | mR@50 / 100 | F@50 / 100 |
| Motif | Vanilla Motif (Zellers et al., 2018) | **36.1 / 37.0** | 13.7 / 14.7 | 19.9 / 21.0 |
| | ST-SGG w/o EMA | 24.6 / 27.4 | 6.0 / 7.7 | 9.6 / 12.0 |
| | ST-SGG w/o $\lambda_c^{inc}, \lambda_c^{dec}$ | 33.0 / 34.4 | 13.5 / 14.3 | 19.2 / 20.2 |
| | ST-SGG | 33.4 / 34.9 | **16.9 / 18.0** | **22.4 / 23.8** |
| BGNN | Vanilla BGNN (Li et al., 2021) | **36.9 / 38.1** | 14.6 / 16.0 | 20.9 / 22.5 |
| | ST-SGG w/o EMA | 36.6 / 37.5 | 13.9 / 14.8 | 20.1 / 21.2 |
| | ST-SGG w/o $\lambda_c^{inc}, \lambda_c^{dec}$ | 33.8 / 36.3 | 10.4 / 12.7 | 15.9 / 18.8 |
| | ST-SGG | 32.3 / 34.0 | 18.0 / 19.0 | 23.1 / 24.4 |
| | ST-SGG+GSL | 33.5 / 34.7 | **18.0 / 19.4** | **23.4 / 24.9** |

inferior performance over all metrics. This degradation is due to the fact that removing class-specific momentum incurs bias to majority classes as shown in Fig. 9 of Appendix E.7. This implies that adjusting the threshold in a class-specific manner, which rapidly/slowly adjusts the threshold for head/tail predicates is important to alleviating the long-tailed problem. When the BGNN backbone is used, similar results are observed. Additionally, we confirmed that ST-SGG+GSL generally outperforms ST-SGG in terms of R@K, mR@K, and F@K on the BGNN backbone, implying that enriching the scene graph structure through GSL helps identify the relations to which pseudo-labels are to be assigned. We further analyze the effect of $\alpha^{inc}$ and $\alpha^{dec}$ Appendix E.8, and Appendix E.9.

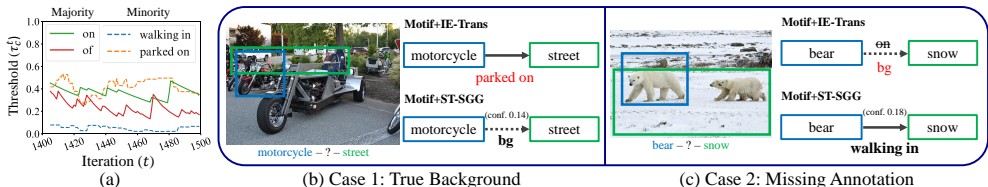

Figure 5: (a) Adaptive threshold values over iterations, and (b) examples of pseudo-labels of ST-SGG and IE-Trans assigned to the different cases. Conf. denotes the confidence, i.e., $\hat{q}$.

## 5.4 QUALITATIVE ANALYSIS ON CATM

We conducted the qualitative analysis to demonstrate the effectiveness of our proposed CATM in assigning accurate pseudo-labels. In Fig 5(a), we observed that the range of the class-specific threshold values defined by CATM is diverse. Interestingly, we found that the threshold is not related to the frequency of classes. This shows the difficulty of determining appropriate class-specific thresholds for pseudo-labeling, and we argue that this is the main reason of the failure of the thresholding techniques used in image recognition in the SGG task. Moreover, let us consider a triplet $\langle \mathsf{motorcycle}, ?, \mathsf{street} \rangle$ shown in Fig. 5(b), which is a case of true background, i.e., it should be labeled with the bg class because the motorcycle in the blue box is far from the street in the green box. We observe that while ST-SGG predicts it correctly as bg, IE-Trans incorrectly predicts it as parked on. This is the due to the design of IE-Trans that tries to assign pseudo-labels to as many unannotated triplets as possible (Zhang et al., 2022). Next, let us consider a triplet $\langle \mathsf{bear}, ?, \mathsf{snow} \rangle$ shown in Fig. 5(c), which is a case of missing annotation, i.e., it should be labeled with a class other than bg because the bear in the blue box is close to the snow in the green box. We observe that IE-Trans initially predicts it as on, and then re-assigns it to bg, due to the behavior of IE-Trans that only pseudo-labels predicates belonging to the minority classes; as on belongs to the majority class, it is re-assigned to bg. On the other hand, ST-SGG predicts it correctly as walking in, which indicates that ST-SGG is a framework that effectively utilizes unannotated triplets by providing accurate pseudo-labels.

## 6 CONCLUSION

Although self-training has shown advancements in computer vision applications like image recognition, its application in SGG has received limited attention. Simply adapting existing self-training methods for SGG is ineffective due to challenges posed by the semantic ambiguity and the long-tailed distribution of predicates. Consequently, we propose a novel self-training framework specifically designed for SGG, called ST-SGG. This framework effectively harnesses unannotated triplets even in the SGG nature, resulting in substantial improvements in fine-grained predicates. Our work offers a new perspective on training SGG models by leveraging existing benchmark scene graph datasets, which often contain missing annotations. For the limitation of this study, please refer to Appendix G.

ACKNOWLEDGEMENTS

This work was supported by Institute of Information & Communications Technology Planning & Evaluation (IITP) grant funded by the Korean government (MSIT) (No. 2020-0-00004, Development of Previsional Intelligence based on Long-term Visual Memory Network, and No.2022-0-00077).

ETHICS STATEMENT

In compliance with the ICLR Code of Ethics, to the best of our knowledge, we have not encountered any ethical issues throughout this paper. Moreover, all the datasets and pre-trained models utilized in our experiments are openly accessible to the public.

REPRODUCIBILITY STATEMENT

To ensure reproducibility of experiment results, we describe the details of datasets, experimental setting, and implementation details in Appendix E and F. Furthermore, we provide a source code in https://github.com/rlqja1107/torch-ST-SGG along with accessible datasets and our pre-trained models.

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

# Supplementary Material

*- Adaptive Self-training Framework for Fine-grained Scene Graph Generation -*

# A    RELATED WORK

**Self-Training.**  Self-training is one of the prominent semi-supervised approaches that utilize a large amount of unlabeled samples, and it has been widely studied in the field of image recognition. The main idea is to assign pseudo-labels to unlabeled samples whose confidence is above a specified threshold and use them for model training to improve the generalization performance of the model. Recent works have focused on reliving the confirmation bias (Arazo et al., 2020) arising from using incorrect pseudo-labels for model training, which in turn undesirably increases the confidence of incorrect predictions. Hence, numerous confidence-based thresholding techniques are proposed to accurately assign pseudo-labels to unlabeled samples. Specifically, UDA (Xie et al., 2020a) and Fixmatch (Sohn et al., 2020) employ a pre-defined constant threshold during training. However, these approaches fail to consider the learning state of a model, aggravating the confirmation bias as the model predicts with higher confidence at the later part of the training step. To resolve the issue, Dash (Xu et al., 2021) presents an adaptive threshold that is decreased as the iteration goes on, and Flexmatch (Zhang et al., 2021) employs a curriculum learning approach to reflect the learning state of the model into the pseudo-labeling process. In addition, Lai et al. (2022); Guo et al. (2021); Wei et al. (2021) assume the situation with an extremely imbalanced class distribution in which the model is prone to a severe confirmation bias. They present thresholding techniques, which change the threshold according to the number of assigned pseudo-labels per class, to reduce the bias towards majority classes. However, none of the above methods addresses the unique challenges of applying a self-training framework in the context of SGG. In this work, we propose a novel thresholding technique that considers the inherent nature of SGG, where the semantic ambiguity of predicates and the bias caused by long-tailed distribution are present.

**Scene Graph Generation.**  Existing SGG methods point out that SGG models accurately predict general predicates, while rarely making correct prediction for fine-grained predicates. There are two mainstream SGG approaches for addressing the biased prediction issue. One line of research aims to alleviate the long-tailed predicate distribution problem. Specifically, Desai et al. (2021); Li et al. (2021) develop triplet-level resampling methods to balance the distribution of general and fine-grained predicates. Yan et al. (2020); Lyu et al. (2022) propose reweighting losses that consider interclass dependencies of predicates to improve the performance on fine-grained predicates. Additionally, Tang et al. (2020) introduces a causal inference framework, and Dong et al. (2022) uses the transfer learning approaches to mitigate the bias towards general predicates. Guo et al. (2021); Chiou et al. (2021) present post-processing methods to address the bias. Most recently, IE-trans (Zhang et al., 2022) proposed to replace general predicates with fine-grained predicates and fill in missing annotations with fine-grained predicates to mitigate the problem caused by the long-tailed predicate distribution. However, IE-trans fails to fully exploit the true labels of unannotated triplets as it uses the initial model parameter of the pretrained model to generate pseudo-labels. On the other hand, we propose a novel pseudo-labeling technique for SGG to effectively utilize the unannotated triplets, which helps reduce the number of incorrect pseudo-labels by iteratively updating both the pseudo-labels and the SGG model.

Another line of research focuses on designing advanced SGG architectures to improve the generalization performance of SGG models. The main idea is to refine the representations of entity and relation proposals obtained from the object detector with message-passing neural networks (MPNN). Specifically, some works enhance the representations by employing sequential models such as RNN and TreeLSTM to capture the visual context (Zellers et al., 2018; Tang et al., 2019; Xu et al., 2017). Rather sequentially capturing the context from neighboring entities, recent works propagate messages from the neighbors using graph neural networks whose representation includes structured context between neighbors (Yang et al., 2018; Lin et al., 2020; Li et al., 2021; Yoon et al., 2023). Relformer (Shit et al., 2022) develops one-stage SGG framework using relational transformer to represent local and global semantic in image as a graph. In this work, we propose a model-agnostic self-training framework for SGG to facilitate the use of many unannotated triplets in the scene graph. We further extend ST-SGG for MPNN-based SGG models based on graph structure learning (GSL), which accurately assigns pseudo-labels based on the presence of relationships.

**Difference from IE-Trans.**  IE-Trans (Zhang et al., 2022) is the most recent work that addresses the inherent issues in scene graph datasets aiming at alleviating the long-tailed problem. Specifically, IE-Trans performs an **internal transfer** that replaces general predicates with fine-grained predicates within the annotated triplets, and an **external transfer** that fills in unannotated triplets, i.e., background classes, with fine-grained predicates. More specifically, IE-Trans utilizes a pretrained SGG

model, such as Motif (Zellers et al., 2018), VCTree (Tang et al., 2019), and BGNN (Li et al., 2021), to generate the confidence for all possible predicates, and sorts them in descending order. Subsequently, it conducts the internal transfer on the top 70% of annotated triplets and external transfer on the top 100% of unannotated triplets (i.e., all unannotated triplets). It is worth noting that IE-Trans assigns pseudo-labels based on the initial parameter of the pre-trained model, and trains a new model from scratch.

On the other hand, ST-SGG focuses only on the effective utilization of unannotated triplets to alleviate the long-tailed problem. That is, we only focus on the "external transfer." In order to accomplish this, ST-SGG learns class-specific thresholds to determine which instances should be pseudo-labeled at every batch, and retraining the model's parameters using the dataset enriched with pseudo-labeled predicates.

Herein, we clarify the difference between ST-SGG and the external transfer of IE-Trans, as they both serve a similar purpose, which is to make use of unannotated triplets.

1. While ST-SGG sets a different threshold for each predicate class, IE-Trans assigns pseudo-labels to predicates based on their confidence values without considering class-specific thresholds. This approach overlooks the diverse distribution of confidence across classes, leading to imprecise pseudo-labeling. For example, in Fig. 6.(a), we observed that the top-1% confidence of carrying is around $0.4$, which is relatively high among confidences of overall predicate classes, while that of painted on below $0.05$, which is relatively low. This implies that when performing external transfer to unannotated triplets sorted in descending order by the confidence, IE-Trans would rarely pseudo-label a triplet with painted on, as its confidence is generally low. We conjecture that, to avoid this problem, IE-Trans performed external transfer on the top 100% of unannotated triplets (i.e., all unannotated triplets) if there is at least a slight overlap between two bounding boxes. In fact, Figure 7.(b) of the IE-Trans paper Zhang et al. (2022) shows that IE-Trans is only successful when the external transfer is performed on the top-100% of unannotated triplets, while the performance degrades significantly otherwise. To make it worse, assigning pseudo-labels to all unannotated triplets leads to the generation of a significant number of incorrect pseudo-labels as the true background relations exist in the scene graphs. On the other hand, ST-SGG addresses these problems by proposing CATM that considers the different confidence distributions across the classes and the presence of true background.

2. ST-SGG is computationally more efficient compared with IE-Trans. ST-SGG only needs to check whether the confidence produced at each batch is greater than the class-specific thresholds to compute the loss for pseudo-labeled triplets. On the other hand, IE-Trans needs to perform an additional data transfer stage, which produces confidence of all possible relationships in the dataset to perform the internal and external transfers. Due to the large number of triplets in the scene graph dataset, this additional stage requires a lot of computational time during the model's inference. This will be further discussed in detail in Appendix E.10.

## B REGARDING CHALLENGES OF APPLYING SELF-TRAINING FOR SGG

### B.1 THRESHOLDING TECHNIQUES IN IMAGE RECOGNITION TASK

As self-training has been widely explored in the field of image recognition, a straightforward approach to designing ST-SGG is to adopt existing thresholding techniques. In the following subsection, we evaluate the existing thresholding techniques (Sohn et al., 2020; Wei et al., 2021; Xu et al., 2021) when they are applied to ST-SGG.

### B.2 DETAILS OF THRESHOLDING TECHNIQUES IN SEC. 3.3

1. **Constant thresholding** (Sohn et al., 2020; Xie et al., 2020a): $\tau = \tau^{\mathrm{con}}$ is a pre-defined constant threshold used in Fixmatch (Sohn et al., 2020). Due to the fact that the SGG model predicts the bg class with significantly higher confidence than the remaining classes, setting a global threshold such as $\tau^{\mathrm{con}} = 0.95$ would fail. Thus, we set the threshold as the top-1% confidence among the confidences for all classes computed on the validation set.

2. **Fixed class-specific thresholding** (Sohn et al., 2020): $\tau = \tau_c^{\mathrm{cls}}$ is a variant of constant thresholding that defines the threshold in a class-specific manner. The threshold for each predicate class $c$ is set to the top-1% confidence per predicate computed on the validation set as in Fig. 6(a).

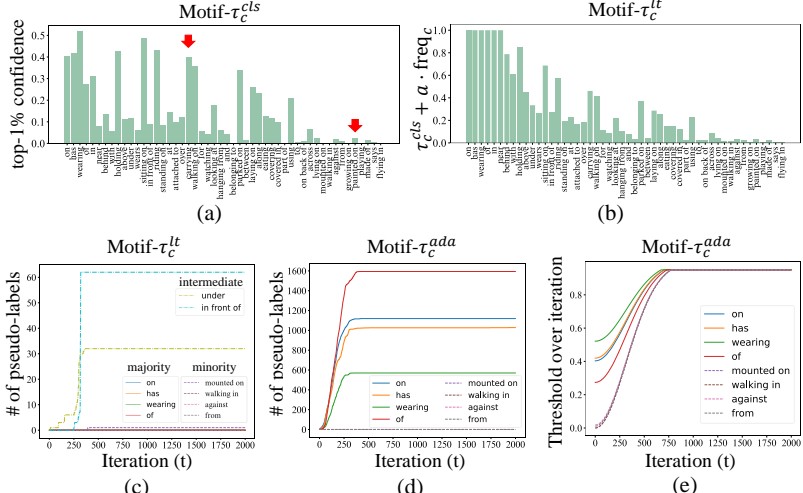

Figure 6: (a),(b) Static thresholds $\tau_c^{\text{cls}}$ and $\tau_c^{\text{lt}}$. (c),(d) The number of pseudo-labels for each predicate, where $\tau_c^{\text{lt}}$ and $\tau_c^{\text{ada}}$ thresholds are used for ST-SGG, respectively. (e) Adaptive threshold over iterations.

3. **Class Frequency-weighted Fixed class-specific thresholding** (Wei et al. (2021)): $\tau = \tau_c^{\text{lt}}$ is designed to consider the nature of long-tailed predicate distribution in SGG to prevent the model from assigning pseudo-labels mainly to majority classes. We set $\tau_c^{\text{lt}}$ as the linear combination of $\tau_c^{\text{cls}}$ and the normalized frequency of predicate class as in Fig. 6(b), which imposes a penalty for assigning pseudo-labels to majority classes.

4. **Class-specific adaptive thresholding** (Xu et al. (2021); Guo & Li (2022)): $\tau = \tau_c^{\text{ada}}$ is a threshold that adaptively changes according to the learning state of the model. Following Dash (Xu et al., 2021), we gradually increase the class-specific threshold starting from $\tau_c^{\text{cls}}$ to reduce the number of incorrect pseudo-labels as the training proceeds (Please refer to Fig. 6(e)).

### B.3 DETAILED DISCUSSION ON RESULT

Here, we delve deeper into the results of Sec 3.3 and explore the reasons why each thresholding technique failed to improve the performance of SGG during self-training.

Despite Motif-$\tau_c^{\text{cls}}$ containing reliable confidence levels for each class (i.e., top-1% confidence per predicate class), we observed that the model only assigns pseudo-labels to predicates belonging to the majority classes (Please see Fig.2(c)), leading to an increased bias towards the majority classes. This indicates that the long-tailed problems must be addressed to design a self-training framework for SGG. To address this, $\tau_c^{\text{lt}}$ was introduced to penalize assigning pseudo-labels with majority predicate classes, but it resulted in a significant performance drop in terms of Recall@100 compared to Motif-$\tau_c^{\text{cls}}$ in Fig. 2(a). This is because, as shown in Fig. 6(c), Motif-$\tau_c^{\text{lt}}$ cannot produce pseudo-labels with majority and minority predicate classes and rather generates pseudo-labels with intermediate classes, leading to the bias towards intermediate classes. It is natural that when the model is biased towards the intermediate classes, the Recall@100 for majority classes decreases since it degrades the generalization performance for majority classes. This result implies that simply penalizing the majority classes is not a solution for providing appropriate thresholds for each predicate class.

On the other hand, $\tau_c^{\text{ada}}$ adjusts the threshold over iterations to control the number of pseudo-labels. However, as depicted in Fig. 6(d), Motif-$\tau_c^{\text{ada}}$ exhibits a bias towards the majority classes in the early iterations and later on, due to the saturated thresholds, it ends up not assigning any pseudo-labels, resulting in performance similar to that of Motif without self-training i.e., Motif-Vanilla. From the experiment, we validated that a new thresholding technique is required for ST-SGG, considering the long-tailed problem and the nature of SGG.

---

**Algorithm 1** ST-SGG

---

**Require:** Pretrained SGG model $f_\theta$, increasing/decreasing rate $\alpha^{inc}, \alpha^{dec}$, coefficient of loss for pseudo-labeled predicates $\beta$, maximum iteration $T$, and sorted number of instances that belongs to each predicate class $N_1, N_2, ..., N_{|\mathcal{C}^p|}$.

1: Compute class-specific momentum: $\lambda_c^{inc} = (\frac{N_c}{N_1})^{\alpha^{inc}}, \lambda_c^{dec} = (\frac{N_{|\mathcal{C}^p|+1-c}}{N_1})^{\alpha^{dec}} \ \forall c \in \mathcal{C}^p$

2: **for** $t = 1$ to $T$ **do**

3:     Given a batch of images $\mathbf{I}_1, ..., \mathbf{I}_B$,

4:         $\mathbf{G}^A = [\mathbf{G}_1^A, \mathbf{G}_2^A, ..., \mathbf{G}_B^A]$ and $\mathbf{G}^U = [\mathbf{G}_1^U, \mathbf{G}_2^U, ..., \mathbf{G}_B^U]$

5:     **for all** $(\mathbf{s}_i, \mathbf{p}_i, \mathbf{o}_i) \in \mathbf{G}^U$ **do**

6:         Compute $\hat{q}_i = \max f_\theta^p(\mathbf{x}_i^{\mathbf{s},\mathbf{o}})$ and assign pseudo-label $\tilde{\mathbf{q}}_i = c$.

7:     **end for**

8:     **if** use GSL **then**

9:         Compute $\hat{s}_i^{\mathbf{s},\mathbf{o}}$ using GSL

10:         $\mathcal{L} = \underbrace{\mathbb{E}_{i \in \mathbf{G}^A}\left[-\mathbf{p}_i \cdot \log(\hat{\mathbf{p}}_i^p)\right]}_{\text{Loss for annotated predicates}} + \underbrace{\mathbb{E}_{i \in \mathbf{G}^U | (\hat{q}_i < \tau_{\tilde{\mathbf{q}}_i}^{t-1}) \vee (\hat{s}_i^{\mathbf{s},\mathbf{o}} < 0.5)}\left[-\mathbf{p}_i^{\mathsf{bg}} \cdot \log(\hat{\mathbf{p}}_i^p)\right]}_{\text{Loss for bg class}}$

11:         $+ \beta \underbrace{\mathbb{E}_{i \in \mathbf{G}^U | (\hat{q}_i \geq \tau_{\tilde{\mathbf{q}}_i}^{t-1}) \wedge (\hat{s}_i^{\mathbf{s},\mathbf{o}} \geq 0.5)}\left[-\tilde{\mathbf{q}}_i \cdot \log(\hat{\mathbf{p}}_i^p)\right]}_{\text{Loss for pseudo-labeled predicates with GSL}}$

12:     **else**

13:         $\mathcal{L} = \underbrace{\mathbb{E}_{i \in \mathbf{G}^A}\left[-\mathbf{p}_i \cdot \log(\hat{\mathbf{p}}_i^p)\right]}_{\text{Loss for annotated predicates}} + \underbrace{\mathbb{E}_{i \in \mathbf{G}^U | \hat{q}_i < \tau_{\tilde{\mathbf{q}}_i}^{t-1}}\left[-\mathbf{p}_i^{\mathsf{bg}} \cdot \log(\hat{\mathbf{p}}_i^p)\right]}_{\text{Loss for bg class}}$

14:         $+ \beta \underbrace{\mathbb{E}_{i \in \mathbf{G}^U | \hat{q}_i \geq \tau_{\tilde{\mathbf{q}}_i}^{t-1}}\left[-\tilde{\mathbf{q}}_i \cdot \log(\hat{\mathbf{p}}_i^p)\right]}_{\text{Loss for pseudo-labeled predicates}}$

15:     **end if**

16:     Update class-specific adaptive threshold          ▷ Refer to Algorithm 2

17:     Train the SGG model $f_\theta$ w/ gradient descent:

18:         $\theta_{t+1} \leftarrow \theta_t - \eta\frac{d\mathcal{L}}{d\theta_t}$

19: **end for**

---

# C    ST-SGG

## C.1    ALGORITHM

For better understanding of ST-SGG, we provide the details of the procedure in Algorithm 1 and Algorithm 2. In Algorithm 1, we process a batch of images by first segregating the batch into an annotated scene graph $\mathbf{G}^A$ and an unannotated scene graph $\mathbf{G}^U$ (Line 4). During each batch iteration, we evaluate confidences and assign pseudo-labels to all the unannotated predicates in $\mathbf{G}^U$ (Line 6). Finally, we calculate three separate losses for the annotated predicates, background predicates, and pseudo-labeled predicates (Line 13). Specifically, for annotated predicates, we employ the standard cross-entropy loss. For unannotated predicates, if their confidence is above the current threshold $\tau_c^t$, we calculate the cross-entropy using the assigned pseudo-label as supervision. If the confidence is below the threshold, the predicate is classified within the background relation class. Notably, we update the thresholds dynamically after each batch, as explained in Algorithm 2.

## C.2    NAIVE MODEL PREDICTION-BASED ADAPTIVE THRESHOLDING

In Sec 4.1, we mentioned that a straightforward approach to estimating the model prediction-based threshold would be to compute the average of the model's confidence in the validation set, and set it as the threshold at every iteration or at a regular iteration interval. Herein, we implement the naive model prediction-based thresholding by setting the class-specific threshold as the averaged confidence computed on the validation set. We computed the confidence with 100 regular intervals since it is impractical to set the class-specific threshold from the validation set at every iteration due to the expensive computation cost. In Fig 7, we observe that the performance of the naive thresholding method significantly degrades between 0 and 100 iterations. We attribute this degradation to the

---

**Algorithm 2** CATM

---

**Require:** Set of confidence $\hat{q}_i$ for $\mathbf{G}^U$, increasing/decreasing rate $\alpha^{inc}, \alpha^{dec}$, and sorted number of instances that belongs to each predicate class $N_1, N_2, ..., N_{|\mathcal{C}^p|}$.

1: **for** $c$ in $\mathcal{C}^p$ **do**
2:     **if** $\exists i \in \mathcal{P}_c^U$ where $\hat{q}_i \geq \tau_c^{t-1}$ **then**
3:         $\tau_c^t = (1 - \lambda_c^{inc}) \cdot \tau_c^{t-1} + \lambda_c^{inc} \cdot \mathbb{E}_{i \in \mathcal{P}_c^U}[\hat{q}_i]$    ($\tau_c^{t-1}$ increases)
4:     **else if** $\hat{q}_i < \tau_c^{t-1}$ for all $i \in \mathcal{P}_c^U$ **then**
5:         $\tau_c^t = (1 - \lambda_c^{dec}) \cdot \tau_c^{t-1} + \lambda_c^{dec} \cdot \mathbb{E}_{i \in \mathcal{P}_c^U}[\hat{q}_i]$    ($\tau_c^{t-1}$ decreases)
6:     **else if** $\mathcal{P}_c^U = \emptyset$ **then**
7:         $\tau_c^t = \tau_c^{t-1}$    ($\tau_c^{t-1}$ remains)
8:     **end if**
9: **end for**
        **return** $[\tau_1^t, ..., \tau_{|\mathcal{C}^p|}^t]$

---

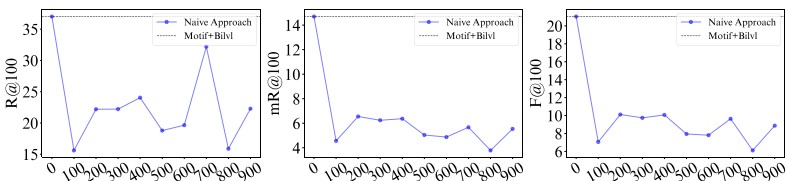

Figure 7: Performance over iterations when naive model prediction-based thresholding is applied on SGCls task.

fact that in SGG it is crucial to reflect the learning state of the model at each iteration, but the naive approach failed to do so. Therefore, the naive approach is inefficient and ineffective compared to the EMA-based approach in terms of reflecting the learning state as well as time complexity.

# D DETAILS ON GRAPH STRUCTURE LEARNER (GSL)

## D.1 GRAPH STRUCTURE LEARNER

For more details of the graph structure learner mentioned in Section 4.3, we formally describe the GSL which is adopted to the MPNN-based SGG model. The main idea of GSL is to enrich the structure of the scene graph by discovering relevant neighbors or removing irrelevant neighbors.

Based on the subject and object representations (i.e., $\mathbf{x^s}, \mathbf{x^o}$), GSL uses an MLP to generate a scalar $s^{\mathbf{s},\mathbf{o}} \in [0,1]$ representing the link probability of the relation between a subject $\mathbf{s}$ and an object $\mathbf{o}$ as follows: $s^{\mathbf{s},\mathbf{o}} = \mathsf{sigmoid}(\mathrm{MLP}([\mathbf{x^s}; \mathbf{x^o}]))$, where [;] is the concatenation operation. A straightforward approach for improving the representation of the relations would be to assign a weight to a message based on the link probability $s^{\mathbf{s},\mathbf{o}}$, so as to treat the messages differently according to their importance. However, as $s^{\mathbf{s},\mathbf{o}}$ is a soft value that lies between 0 and 1, this approach fails to completely prevent irrelevent messages from being propagated among entities, i.e., this approach essentially treats a graph as a fully connected graph where the edge weights are between 0 and 1. To this end, we propose to sample relevant relations from a scene graph based on $s^{\mathbf{s},\mathbf{o}}$ so that messages are only allowed to be propagated through the sampled relations. However, as sampling is a discrete process that is not differentiable, which hinders the end-to-end training of our model, we apply the Gumbel-softmax reparameterization trick (Maddison et al., 2016; Jang et al., 2016) as $\hat{s}^{\mathbf{s},\mathbf{o}} = \mathrm{Bernoulli}\left[\frac{1}{1+\exp(-(\log s^{\mathbf{s},\mathbf{o}}+\varepsilon)/\tau)}\right]$, where $\mathrm{Bernoulli}(\cdot)$ is the Bernoulli approximation, $\varepsilon \sim \mathrm{Gumbel}(0,1)$ is the Gumbel noise for reparameterization, and $\tau$ is the temperature hyperparameter. In the forward pass, we sample the relation between $\mathbf{s}$ and $\mathbf{o}$, if $\hat{s}^{\mathbf{s},\mathbf{o}} > 0.5$, while in the backward pass, we employ the straight-through gradient estimator (Bengio et al., 2013) so that the gradient can be passed through the relaxed $s^{\mathbf{s},\mathbf{o}}$. That is, only for $\mathbf{s}$ and $\mathbf{o}$ with $\hat{s}^{\mathbf{s},\mathbf{o}} > 0.5$, we allow message passing between $\mathbf{s}$ and $\mathbf{o}$, and consider the relation between between $\mathbf{s}$ and $\mathbf{o}$ as a candidate for pseudo-labeling.

## D.2 TRAINING

We train the GSL by optimizing the following binary classification loss:

$$\mathcal{L}_{GSL} = -\sum_{b=1}^{B}\sum_{k=1}^{|\mathbf{G}_b|} y^{\mathbf{s}_k,\mathbf{o}_k}(1-s^{\mathbf{s}_k,\mathbf{o}_k})^{\gamma} \cdot \log(s^{\mathbf{s}_k,\mathbf{o}_k}) \tag{3}$$

where $y^{\mathbf{s}_k,\mathbf{o}_k}$ is the binary value that equals to 1 if a relation exists between subject $\mathbf{s}_k$ and object $\mathbf{o}_k$, and otherwise 0 (i.e., when bg), and $\gamma$ is a hyperparameter. As the number of bg greatly outnumbers that of the remaining classes, i.e., class imbalance, we use the focal loss (Lin et al., 2017) instead of the binary cross-entropy loss.

# E EXPERIMENT ON VISUAL GENOME

## E.1 DATASET

We follow the commonly used pre-processing strategies that have been extensively employed for evaluating SGG (Yoon et al., 2023; Li et al., 2021; Zellers et al., 2018; Xu et al., 2017). The Visual Genome dataset comprising 108K images is divided into a 70% training set and a 30% test set, with 5K images from the training set utilized for the validation set. Based on the frequency of occurrence, we only consider the top 150 most frequently occurring object classes and the top 50 predicate classes. Following the pre-processing steps, the average number of objects per image is 11.6, while the average number of predicates is 6.2.

## E.2 EXPERIMENTAL SETTING

**Evaluation Metric.** We evaluate SGG models on three metrics: (1) Recall@K (R@K) is the conventional metric, which calculates the proportion of top-K predicted triplets that are in ground truth. (2) mean Recall@K (mR@K) calculates the average of the recall for each predicate class, which is designed to measure the performance of SGG models under the long-tailed predicate class distribution. (3) F@K calculates the harmonic average of R@K and mR@K to jointly consider R@K and mR@K without trade-off between these metrics. Recent SGG studies (Li et al., 2021; Zhang et al., 2022) addressing the long-tailed distribution focus on enhancing the performance of the minority predicate classes (measured by mean R@K) since they usually include more informative descriptions in depicting a scene (e.g., "walking in", "playing"), than the majority classes (e.g., "on", "has") does. However, there is a trade-off between R@K and mR@K. In other words, if a model deliberately lowers the number of predictions for head predicate classes (e.g., "on") while increasing it for tail predicate classes (e.g., "standing on", "walking on", and "walking in"), we would encounter a decrease in R@K and an increase in mR@K. That being said, we can deliberately increase R@K at the expense of reduced mR@K and vice versa. Thus, we focus on enhancing the F@K, considering the trade-off between Recall@K and mean Recall@K.

**Evaluation Protocol.** We evaluate under three conventional SGG tasks : (1) Predicate Classification (PredCls) provides the ground truth bounding box and the class of entities, and then evaluate the performance of SGG models in terms of recognizing the predicate class. (2) Scene Graph Classification (SGCls) only provides the bounding box, and requires the model to predict the class of entities and predicates between them. (3) In Scene Graph Detection (SGDet), SGG models generate entity proposals, and predict the classes of entities and predicates between them.

**Baselines.** We include the baselines that can be classified into 1) **model-agnostic** framework and 2) **specific** models. For the model-agnostic framework, we compare ST-SGG with unbiased SGG models such as **TDE** (Tang et al., 2020), **DLFE** (Chiou et al., 2021), **Re-sampling** (Li et al., 2021), **NICE** (Li et al., 2022), and **IE-Trans** (Zhang et al., 2022). Moreover, we also include the variant of IE-Trans ( i.e., **I-Trans**), which excludes the pseudo-labeling mechanism (i.e., E-Trans) on unannotated triplets, to further compare the effectiveness of its pseudo-labeling with ST-SGG. For the specific model, we include the state-of-the-art models such as DT2-ACBS (Desai et al., 2021), PCPL (Yan et al., 2020), KERN (Chen et al., 2019), and GBNet (Zareian et al., 2020).

### E.3 Implementation Details

Following previous studies (Tang et al., 2020; Yoon et al., 2023; Li et al., 2021; Zhang et al., 2022), we employ Faster R-CNN (Ren et al., 2015) with ResNeXt-101-FPN (Xie et al., 2017) backbone network as the object detector, whose pretrained parameters are frozen while training the SGG model. In SGDet task, we select the top 80 entity proposals sorted by scores computed by object detector, and use per-class non-maximal suppression (NMS) at IoU 0.5. For ST-SGG, we conduct a grid search for the rate in momentum $\alpha^{inc}$ and $\alpha^{dec}$ with an interval of 0.2 (Sec. 4.2), and set the coefficient of the loss for pseudo-labeled predicates $\beta$ to 1.0 (Equation 1) in the base SGG model. On the other hand, we set the $\beta$ to 0.1 when adopting the re-weight loss in Appendix E.4. Following the self-training framework, we initially train the SGG model based on the annotated triplets and then re-train it using both annotated triplets and pseudo-labeled triplets. We set the initial value of $\tau_c^{(t)}$ for all predicate classes to 0 based on our observation that the performance of ST-SGG is not sensitive to the initial value. We set the maximum number of pseudo-labeled instances per class to 3 in an image. Similar to IE-Trans, we give pseudo-labels on unannotated triplets only when the bounding boxes of the subject and object overlap. For the graph structure learner, we set the temperature $\tau$ to 0.5 and $\gamma$ in focal loss to 2.0. For each experiment, we used the A6000 GPU device.

Table 4: Performance (%) comparison of ST-SGG with Reweight loss (Rwt) approach on three tasks. The best performance is shown in bold.

| Model | PredCls | | | SGCls | | | SGDet | | |
|---|---|---|---|---|---|---|---|---|---|
| | R@50 / 100 | mR@50 / 100 | F@50 / 100 | R@50 / 100 | mR@50 / 100 | F@50 / 100 | R@50 / 100 | mR@50 / 100 | F@50 / 100 |
| Motif (Zellers et al., 2018)+Rwt | 57.5 / 60.0 | 30.1 / 32.6 | 39.5 / 42.2 | 34.1 / 35.2 | 16.8 / 17.8 | 22.5 / 23.6 | 26.4 / 30.9 | 14.1 / 16.4 | 18.4 / 21.4 |
| +ST-SGG | 49.5 / 52.0 | 33.8 / 36.2 | 40.2 / 42.7 | 31.1 / 32.4 | 17.9 / 19.3 | 22.7 / 24.2 | 23.4 / 27.8 | 14.7 / 17.3 | 18.1 / 21.3 |
| +I-Trans | 49.5 / 51.5 | 36.0 / 39.0 | 41.7 / 44.4 | 28.0 / 28.9 | 20.5 / 21.9 | 23.7 / 24.9 | 23.2 / 27.1 | 14.7 / 17.3 | 18.0 / 21.1 |
| +IE-Trans | 48.6 / 50.5 | 35.8 / 39.1 | 41.2 / 44.1 | 29.4 / 30.3 | 21.0 / 22.2 | 24.5 / 25.6 | 23.4 / 27.2 | 14.9 / 17.5 | 18.2 / 21.3 |
| +I-Trans+ST-SGG | 48.1 / 50.0 | 38.0 / 40.7 | 42.5 / 44.9 | 27.3 / 28.1 | 22.0 / 23.3 | 24.4 / 25.5 | 21.4 / 25.2 | 16.0 / 19.2 | 18.3 / 21.8 |
| BGNN (Li et al., 2021)+Rwt | 57.2 / 59.7 | 25.7 / 28.1 | 35.5 / 38.2 | 34.5 / 35.8 | 15.1 / 16.1 | 21.0 / 22.2 | 25.8 / 29.3 | 10.4 / 12.2 | 14.8 / 17.2 |
| +ST-SGG | 53.2 / 55.5 | 29.3 / 32.1 | 37.8 / 40.7 | 32.7 / 33.9 | 16.7 / 18.2 | 22.1 / 23.7 | 23.6 / 27.1 | 11.4 / 14.1 | 15.4 / 18.5 |
| +I-Trans | 48.8 / 50.7 | 35.5 / 38.5 | 41.1 / 43.8 | 29.9 / 30.7 | 20.6 / 21.8 | 24.4 / 25.5 | 21.8 / 25.1 | 13.8 / 16.8 | 16.9 / 20.1 |
| +IE-Trans | 48.5 / 50.4 | 36.5 / 39.3 | 41.7 / 44.2 | 29.6 / 30.6 | 20.5 / 21.7 | 24.2 / 25.4 | 21.2 / 24.5 | 14.2 / 16.7 | 17.0 / 19.9 |
| +I-Trans+ST-SGG | 47.9 / 49.7 | 37.2 / 40.1 | 41.9 / 44.4 | 29.1 / 30.0 | 21.1 / 22.6 | 24.5 / 25.8 | 20.1 / 23.5 | 15.1 / 18.2 | 17.3 / 20.5 |

### E.4 Result with Reweighting Methods

In addition to the experiments with re-sampling (Li et al., 2021) and I-Trans (Zhang et al., 2022) approach, which is presented in Table 1 of the main paper, we further conduct experiments for ST-SGG with a conventional re-weight loss (Rwt) and show the results in Table 4. Note that we follow the conventional re-weight loss used in IE-Trans (Zhang et al., 2022). We observe that Motif/BGNN+Rwt+ST-SGG generally improves the performance on mR@K and F@K compared to Motif/BGNN+Rwt. Furthermore, Motif/BGNN+Rwt+I-Trans+ST-SGG achieves even better performance. These findings, consistent with the insights discussed in Section 5.1, support that ST-SGG with a simple debiasing method (i.e., re-weight) shows the effectiveness of utilizing unannotated triplets, resulting in further alleviating the long-tailed problem.

### E.5 Zero-shot Performance Analysis

Table 5: Zero-shot (zR) Performance when Motif and VCTree backbones are used.

| Method | zR@50 / 100 | Method | zR@50 / 100 |
|---|---|---|---|
| Motif | 16.3 / 19.4 | VCTree | 16.7 / 19.7 |
| Motif+Resampling | 17.0 / 19.5 | VCTree+Resampling | 16.3 / 19.4 |
| Motif+IE-Trans | 13.0 / 15.9 | VCTree+IE-Trans | 14.0 / 16.5 |
| Motif+ST-SGG | 18.0 / 21.0 | VCTree+ST-SGG | 19.8 / 21.0 |

Following the zero-shot settings (Tang et al., 2020), we compute the recall@K values for specific (subject, predicate, object) pairs, which are not included in the training set but emerge in the testing phase. Table 5 reveals that debiasing techniques, including resampling and IE-Trans, do not enhance the zero-shot performance of SGG models. To elaborate, the limitation of the resampling approach lies in its simplistic alteration of predicate class frequencies, which does not augment any variance in the data. As for IE-Trans, it generates pseudo-labels only for minority predicates classes, which restricts the ability of finding zero-shot triplets. Conversely, the application of ST-SGG shows improvements in the zero-shot performance for both Motif and VCTree models. This improvement suggests that ST-SGG can generate pseudo-labels that are instrumental in enhancing the model's

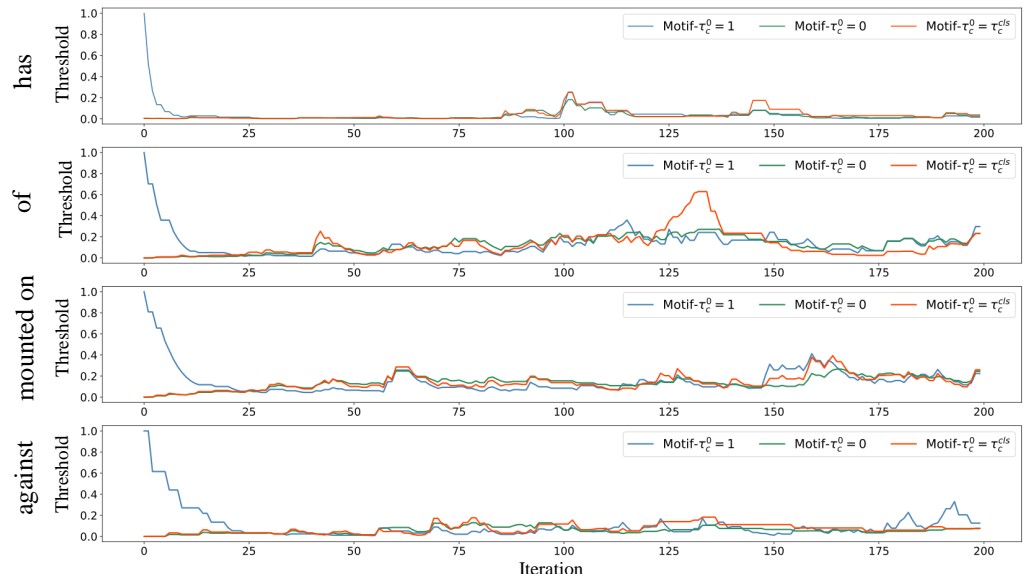

Figure 8: Adaptive threshold values over iterations on SGCls task when $\tau_c^0=1$, $\tau_c^0=0$, and $\tau_c^0 = \tau_c^{cls}$. Motif backbone is used.

generalization to unseen triplet patterns. The effectiveness of ST-SGG in this context underscores its potential as a robust tool for improving the generalization capabilities of SGG models, where the ability to predict novel patterns is crucial.

### E.6 EFFECT OF INITIAL VALUE FOR CLASS-SPECIFIC THRESHOLD

We set the initial value of $\tau_c$ for all predicate classes to 0 based on our observation that the performance of ST-SGG is not sensitive to the initial value. We further showcase the effect of the initial value of the class-specific adaptive threshold. Specifically, we train Motif with bi-level sampling, and consider three cases: initial thresholds for all predicates are one (i.e., Motif-$\tau_c^0 = 1$), zero (i.e., Motif-$\tau_c^0 = 0$), and the mean value of the confidence computed on the validation set (i.e., Motif-$\tau_c^0$=Motif-$\tau_c^{cls}$ in the main paper.). In Table 6, we observe that the performance of ST-SGG is not sensitive to the selection of the initial threshold in terms of F@K. Moreover, in Fig. 8, CATM eventually adjusts thresholds for all predicates in a narrow range. These results imply that CATM rapidly finds the proper threshold wherever they started from and, in turn, makes the value of $\tau_c$ stabilize within a few iterations.

Table 6: Performance comparison when $\tau_c^0 = 1$, $\tau_c^0 = 0$, and $\tau_c^0 = \tau_c^{cls}$. Motif backbone is used.

| Method | SGCls task | | |
|---|---|---|---|
| | R@50 / 100 | mR@50 / 100 | F@50 / 100 |
| Motif+Resampling | 36.1 / 37.0 | 13.7 / 14.7 | 19.9 / 21.0 |
| ST-SGG w/ $\tau_c^0$=1 | 32.2 / 33.6 | 17.3 / 18.5 | 22.5 / 23.9 |
| ST-SGG w/ $\tau_c^0$=0 | 33.4 / 34.9 | 16.9 / 18.0 | 22.4 / 23.7 |
| ST-SGG w/ $\tau_c^0 = \tau_c^{cls}$ | 31.9 / 33.6 | 16.5 / 18.1 | 21.8 / 23.5 |

### E.7 EFFECT OF CLASS-SPECIFIC MOMENTUM

We conducted an analysis on the effect of class-specific momentum which is described in Section 4.2 of the main paper. Fig. 9(a) shows that the SGG model without the class-specific momentum produces a significantly more pseudo-labels for majority predicates compared to minority predicates, exacerbating the long-tailed problem, which leads to a decrease in performance, as shown in Table 3 of the main paper. On the other hand, Fig. 9(b) demonstrates that unannotated triplets are pseudo-

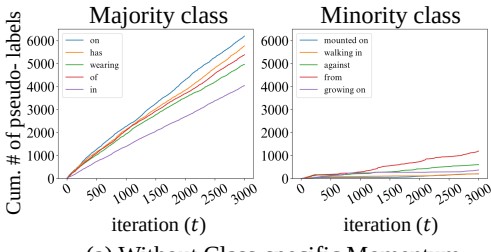 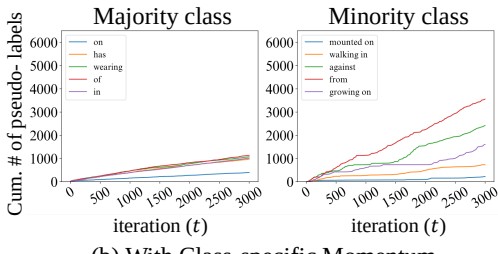

(a) Without Class-specific Momentum  (b) With Class-specific Momentum

Figure 9: The cumulative number of assigned pseudo-labels of several predicate classes in the majority and minority classes (a) without class-specific momentum (i.e., $\lambda^{inc} = \lambda^{dec} = 0.5$) and (b) with class-specific momentum. Unannotated triplets are mainly pseudo-labeled with majority (i.e., head) predicate classes in (a), whereas minority (i.e., tail) predicate classes are assigned more in (b).

labeled more with predicates from minority classes compared to majority classes. This demonstrates that applying *class-specific momentum* relieves the long-tailed problem by generating the pseudo-labels annotated with minority predicates classes.

## E.8 EFFECT OF $\alpha^{inc}$ AND $\alpha^{dec}$

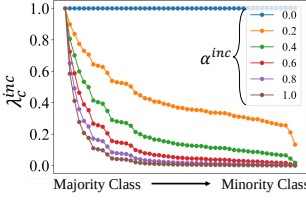 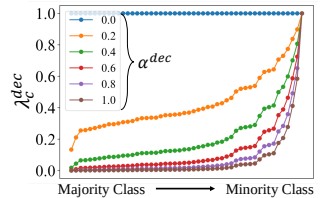 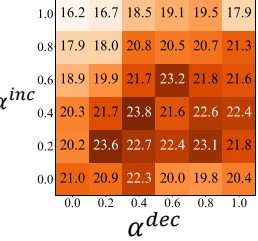

Figure 10: $\lambda_c^{inc}$ and $\lambda_c^{dec}$ according to the change of $\alpha^{inc}$ and $\alpha^{dec}$, respectively, which is represented by lines' color.

Figure 11: Effect of $\alpha^{inc}$ and $\alpha^{dec}$ with F@100 performance.

Fig. 11 shows the performance (F@100) according to the change of $\alpha^{inc}$ (y-axis) and $\alpha^{dec}$ (x-axis), which control the increasing and decreasing rate in EMA, respectively. Note that the smaller the value of $\alpha^{inc}$ or $\alpha^{dec}$ is, the more aggressive the threshold of minority classes increases or decreases, which is described in detail in Fig. 10. We observe that a high value of $\alpha^{inc}$ (row 1, 2) leads to a decrease in performance since a high value of $\alpha^{inc}$ severely restricts the increase of the threshold for minority classes, resulting in incorrect pseudo-label assignment for minority classes. Compared to the performance on the high value of $\alpha^{inc}$, the high value of $\alpha^{dec}$ shows relatively competitive performance. This suggests that a gradual decrease in the threshold for minority classes is acceptable since it keeps assigning the confident pseudo-labels. However, the best performance is achieved with $\alpha^{inc} = 0.4$ and $\alpha^{dec} = 0.4$, rather than $\alpha^{dec} = 1.0$ or $0.8$, indicating that in the self-training for SGG task, it is beneficial to appropriately increase or decrease the threshold while assigning more pseudo-labels on minority classes.

## E.9 EFFECT OF $\beta$

Fig. 12 shows the performance with respect to the coefficient of the loss for the pseudo-labeled predicates i.e., $\beta$ in Equation 1. Here, we also employ the Motif backbone network trained with bi-level resampling techniques (Li et al., 2021). We observed consistent performance of ST-SGG ranging from 0.1 to 1.0 for the coefficient $\beta$, which indicates that ST-SGG is not sensitive to the coefficient $\beta$ of the loss for the pseudo-labeled predicates. Moreover, we noticed a performance drop when $\beta$ becomes too large, suggesting that it is preferable to prioritize the existing annotated predicates over the pseudo-labeled predicates to train SGG models.

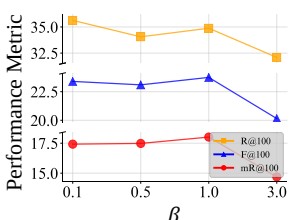

Figure 12: Sensitivity of $\beta$ in Equation 1

### E.10 COMPLEXITY ANALYSIS

Table 7 shows the total training time of Motif-Vanilla, IE-Trans, and ST-SGG required for 20,000 training iterations. Note that the 20,000 iterations include "Pseudo-labeling," and "Re-training." We observed that ST-SGG requires a shorter training time than IE-Trans for pseudo-labeling and re-training. This implies that the iterative pseudo-labeling process of ST-SGG is computationally more efficient than the pseudo-labeling process adopted by IE-Trans.

Table 7: Training time of Motif-backbone models during 20,000 iterations with batch size 6. A6000 GPU device is used.

| Model | Trainig time on PredCls task (20,000 iterations) | | | |
|---|---|---|---|---|
| | Pre-training | Self-training | | Total |
| | | Pseudo-labeling | Re-training | |
| Vanilla | 2h 21m | - | - | 2h 21m |
| IE-Trans | | 1h 39m | 2h 21m | 6h 21m |
| ST-SGG | | 2h 42m | | 5h 03m |

Please note that ST-SGG utilizes the confidence originally produced by the backbone SGG model during the training stage, resulting in very low additional computational cost. For example, when Motif-vanilla generates the confidence to compute the loss for annotated triplets, ST-SGG compares it with the learned thresholds and determines whether to include it in the loss for pseudo-labeled predicates. In other words, ST-SGG only requires additional computations for calculating the gradients of the loss for pseudo-labeled predicates. On the other hand, IE-Trans has additional inference stage after training Motif-Vanilla, and stores confidence for every entity pair in the dataset to obtain the rank of all confidence.

## F EXPERIMENT ON OPEN IMAGE V6

### F.1 DATASET

We closely follow the data processing of previous works (Li et al., 2021; Yoon et al., 2023) for Open Image V6 (OI-V6). After preprocessing, OI-V6 is split into 126,368 train images, 1,813 validation images, and 6,322 test images, and contains 301 object classes, and 31 predicate classes.

### F.2 EXPERIMENTAL SETTING

**Evaluation Metric.** Following (Zhang et al., 2019), we additionally evaluate SGG models on three metrics: $\text{wmAP}_{rel}$ is weighted mean average precision of relationships, $\text{wmAP}_{phr}$ is weighted mean average precision of phrase, and the final score is computed by $\text{score}_{\text{wtd}} = 0.2 \times R@50 + 0.4 \times \text{wmAP}_{rel} + 0.4 \times \text{wmAP}_{phr}$. Specifically, $\text{wmAP}_{rel}$ evaluates the average precision of subject, predicate and object, where both the ground truth boxes of subject and object have an IOU greater than 0.5 with the ground truth bounding boxes, and compute the weighted sum of the average precision scaled with each frequency of predicate class. $\text{wmAP}_{phr}$ is similarly computed to $\text{wmAP}_{rel}$, but it computes single bounding box enclosing both subject and object with IOU greater than 0.5.

**Evaluation Protocol.** Following the previous works (Li et al., 2021; Lin et al., 2020; Zhang et al., 2019), we evaluate on Scene Graph Detection (SGDet) task, where SGG models generate entity and relation proposals, and predict the classes of entities and the class of predicates between them.

### F.3 COMPARISON WITH THE STATE-OF-ART MODELS

Table 8 shows the result in SGDet task on Open ImageV6 datasets. We have the following observations: 1) ST-SGG generally improves Motif, Motif with the resampling, and BGNN in terms of mR@50 and F@50, implying that the long-tailed problem is alleviated by our pseudo-labeling method. 2) Interestingly, Motif+ST-SGG shows competitive performance among those with Motif backbone in terms of R@50 and wmAP. It is important to note that R@50 and wmAP[1] metrics primarily emphasize the performance of head predicates, which is in contrast to the objective of addressing the long-tailed problem. This implies that our proposed framework retains the performance on head predicates while achieving the state-of-art performance on tail predicates. This implication

---

[1]The wmAP metric is computed by $\text{AP}_{rel} \times \text{weight}$, where the weight is based on the frequency in the test data.

Table 8: Results on Open Images V6 (Kuznetsova et al., 2020). **Bold** represents the best performance, and underline represents the second best.

| Model | R@50 | mR@50 | F@50 | wmAP$_{rel}$ | wmAP$_{phr}$ | score$_{wtd}$ |
|---|---|---|---|---|---|---|
| RelDN Zhang et al. (2019) | **75.3** | 37.2 | 49.8 | 32.2 | 33.4 | **42.0** |
| G-RCNN Yang et al. (2018) | 74.5 | 34.0 | 46.7 | 33.2 | 34.2 | 41.8 |
| GPS-Net Lin et al. (2020) | 74.7 | 38.9 | 51.2 | 32.8 | 33.9 | 41.6 |
| Motif Zellers et al. (2018) | 70.6 | 31.8 | 43.8 | 30.3 | 31.1 | 38.5 |
| +ST-SGG | 71.8 | 34.1 | 46.2 | 31.1 | 32.5 | 39.8 |
| +TDE Tang et al. (2020) | 69.3 | 35.5 | 46.9 | 30.7 | 32.8 | 39.3 |
| +Resamp. Li et al. (2021) | 70.3 | 40.6 | 51.5 | 32.0 | 34.0 | 40.2 |
| +Resamp.+ST-SGG | 71.7 | 42.7 | 53.5 | **32.9** | **35.2** | 41.4 |
| BGNN Li et al. (2021) | **75.3** | 39.8 | 52.1 | **32.9** | 34.5 | 41.9 |
| +ST-SGG | 75.2 | **42.9** | **54.6** | 32.8 | 34.2 | 41.7 |

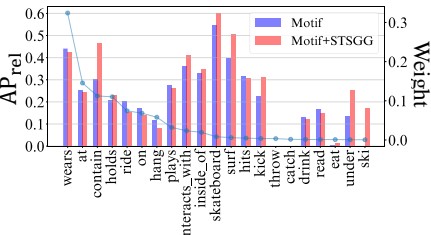

Figure 13: AP$_{rel}$ performance comparison per class. The blue line and *Weight* (i.e., right $y$-axis) represent the frequency ratio in the test data.

can be further explored by referring to Fig. 13, which displays the performance AP$_{rel}$ for each class. Specifically, for the head predicates, Motif+ST-SGG achieves competitive results, particularly in the case of the contain predicate. For the tail predicates, Motif+ST-SGG significantly enhances the performance, particularly in the cases where Motif struggles to make accurate predictions, such as ski predicates.

## G  LIMITATION & FUTURE WORK

Although our current work (including existing SGG models) focuses on utilizing benchmark scene graph datasets, ST-SGG has a potential to leverage external resources. For instance, ST-SGG can benefit from leveraging localization datasets (Lin et al., 2014) containing only the bounding box information without annotation. Specifically, we can include these unannotated bounding boxes obtained from localization datasets in the second and third terms in the loss (Equation 1), i.e., add them to $\mathbf{G}_b^U$. This approach is promising in utilizing resources to enrich the benchmark scene graph datasets, where acquiring annotated triplets is costly.

