# OpenReview forum: "Adaptive Self-training Framework for Fine-grained Scene Graph Generation"
_ICLR.cc/2024/Conference — ICLR 2024 poster_

### Official Review · Reviewer_QGUt · 2023-10-25

**Soundness:** 2 fair
**Presentation:** 3 good
**Contribution:** 2 fair
**Rating:** 5
**Confidence:** 4

**Summary:**

This paper proposes a self-training framework for the SGG task, which uses the unannotated triplets for imbalanced SGG. With the class-specific adaptive thresholding with momentum algorithm, their model can obtain some pseudo labels to deal with the missing annotations problem. Finally, they verify the effectiveness of ST-SGG on VG and OI-V6 datasets.

**Strengths:**

1. The idea is well-motivated and intuitive. They provide an in-depth analysis of long-tailed SGG.
2. The paper is well written and easy to understand.
3. Extensive experiments and ablations verify the effectiveness of enhancing the performance of tail classes.

**Weaknesses:**

1.The technical contribution of this paper might be limited. The self-training framework is used for image classification, the adaptive thresholding is used to obtain accurate pseudo labels in many tasks, and the GSL is also not new. The authors should take more discussion about their novelty and variant in SGG.

2.There are some SOTA SGG methods which might get higher performance than this paper. These methods should also be included for comparisons and analyses, such as VETO[1], GCL[2], PE-Net-Reweigh[3], and so on.
[1] Vision Relation Transformer for Unbiased Scene Graph Generation, ICCV2023.
[2] Stacked Hybrid-Attention and Group Collaborative Learning for Unbiased Scene Graph Generation, CVPR 2022.
[3] Prototype-based Embedding Network for Scene Graph Generation, CVPR 2023.

**Questions:**

See Strengths and Weaknesses.

---

> ### Author Response · Authors · 2023-11-16
>
> **(W1) More Discussion about Novelty and Variant in SGG**
>
> Thanks for pointing out an issue regarding the technical novelty of our paper. We fully agree that incorporating a discussion regarding the novelty and the variant in SGG will enhance the quality of our research. Beyond the novelty arising from the unique challenges in applying self-training to SGG mentioned in the paper, i.e., semantic ambiguity and the long-tailed problem, making it difficult to determine an appropriate threshold for each predicate class, we would like to emphasize the novelty of our work in the following three aspects mentioned by the reviewer:
>
> ----
>
> **1. Difference from Self-training in Image recognition.**
>
> We agree with the reviewer that the self-training framework has been used for image classification. However, in this work, we identify and thoroughly analyze the failure of conventional self-training methods developed for image classification when they are applied to SGG models (Please refer to Section 3.3). Despite self-training being a well-explored technique, its application in the context of SGG was previously unfeasible. More specifically, while the self-training in image classification has addressed imbalanced class situations or determined dynamic thresholds based on the model's learning state, these issues are not independent but intermingled when it comes to the SGG task, requiring a combined approach to handle the issues. Additionally, the presence of a ‘bg’ class, which indicates the absence of a relation and is prevalent in 95.5% of object pairs, complicates the determination of an appropriate threshold. Traditional self-training methods, not considering these factors, often degraded the model's performance, as demonstrated in Sec 3.3 of our paper. To resolve the above issues together, we propose to adopt the EMA technique with class-specific momentum. We believe that our study has a technical contribution in that we consider the inherent nature of SGG to adopt the self-training framework to SGG models.
>
> To further strengthen our claim, we would like to elaborate on one of our experiments. In Table 3 of the main paper, our ablation study shows that excluding any component of CATM results in performance inferior to the Vanilla backbone model. This highlights the significance of simultaneously considering semantic ambiguity, the long-tailed problem, and the presence of the bg class when applying self-training to SGG.
>
> Once again, we agree that self-training has been continuously researched in the field of image recognition. However, despite numerous studies, its application has been very limited in the SGG task. This limitation, we believe, is due to the unique challenges in SGG, i.e., semantic ambiguity of predicate classes and long-tailed predicate distribution (as mentioned in Section 1), that have hindered the application of self-training methods. Therefore, we consider our work, which for the first time creates a connection between the broad domain of self-training and SGG, to be a significant contribution. This connection will enable the potential development of the SGG field by adapting various self-training techniques.
>
> ----
>
>
> **2. Regarding Adaptive Thresholding Used in Many Tasks**
>
> We agree with the reviewer that the adaptive thresholding has been used to obtain accurate pseudo-labels in many tasks. Hence, in Section 3.3, we checked whether a recently proposed class-specific adaptive thresholding technique [1], which increases the threshold as the iteration proceeds, works for the SGG task (please refer to Motif-$\tau_c^{ada}$ in Figure 2(a)). However, we observed that incorrect pseudo-labels are assigned in the early training steps while the threshold saturates in the later training steps, which prevents the model from further assigning any pseudo-labels. This implies that a naive implementation of the class-specific adaptive thresholding technique fails to fully exploit the unannotated triplets in self-training for SGG. To this end, we devised a novel pseudo-labeling technique for SGG, which adaptively adjusts the class-specific threshold by considering not only the long-tailed predicate distribution but also the model’s learning state of each predicate class.
>
> [1] Dash: Semi-Supervised Learning with Dynamic Thresholding. Yi Xu et al. ICML’21.

---

> ### Author Response · Authors · 2023-11-16
>
> ----
>
>
> **3. Regarding the Graph Structure Learner.**
>
> We agree with the reviewer that GSL is not new and it has been previously used in MPNN-based methods [1,2] to prune dense graphs. Specifically, Graph R-CNN uses a score computed by an MLP to select K edges with high scores [1]. BGNN uses the score to gate the message-passing between objects and relations [2]. However, the uniqueness of the GSL in our framework lies in its synergy with the self-training technique. Specifically, we discovered that it is especially helpful to only consider the relevant relations predicted by GSL as candidates for pseudo-labeling. Moreover, as mentioned in Section 4.3, adopting GSL helps generate relatively low confident predictions on the “bg” class, allowing the model to make more confident predictions on the remaining predicate classes (See Figure 3 of the paper), which is particularly beneficial when setting the class-specific thresholds for pseudo-labeling unannotated triplets. In summary, while the GSL used in traditional MPNNs [1,2] is only used to determine the edges in which messages are to be propagated, the GSL used in our work is further utilized to determine the candidates for pseudo-labeling.
>
> Furthermore, we would like to draw attention again to the experimental result on the effect of GSL in our paper. From Table 2 (row 3, 6) in the main paper, we observed that using GSL for determining candidate relations for pseudo-labeling improved the performance of BGNN+ST-SGG. It implies that using the GSL score for pseudo-labeling is an effective strategy to enhance the performance of the SGG model.
>
> [1] Jianwei Yang, Jiasen Lu, Stefan Lee, Dhruv Batra, and Devi Parikh. Graph r-cnn for scene graph generation. ECCV 2018.
> [2] Rongjie Li, Songyang Zhang, Bo Wan, and Xuming He. Bipartite graph network with adaptive message passing for unbiased scene graph generation. CVPR, 2021.
>
> ----
>
> **Last but not least, we would emphasize another contribution of ST-SGG in the perspective of data efficiency.** Note that the SGG task usually requires annotations of images with bounding boxes, object classes, and relation classes, ensuring that they are accurately matched. Hence, compared with other tasks, e.g., the image classification, the cost of data annotation for SGG is more expensive. Moreover, human annotations may be noisy. In this regard, we argue that ST-SGG, which is a pioneering self-training framework for SGG, is particularly meaningful. We believe that our research can substantially reduce the data annotation cost by utilizing the missing annotations, or unannotated dataset based on ST-SGG.
>
> We would like to express our sincere gratitude to the reviewer once again for suggesting the point that strengthens the advantage of our work.

---

> ### Author Response · Authors · 2023-11-16
>
> **(W2) Experiments with more state-of-the-art baselines**
>
> We express our appreciation to the reviewer for mentioning studies [1, 2, 3] that we had missed.
>
> - **Comparison with [2]:** We compared with [2], which utilizes the group collaborative learning (GCL). We updated Table 1 of the main paper in **the modified PDF** to include a comparison with GCL. We observed that GCL significantly improves mR@K, whereas it sacrifices a considerable amount of R@K. Hence, our proposed ST-SGG still surpasses GCL in terms of F@K metrics, which implies a great balanced performance on the overall predicate and tail predicates of our method.
>
> - **Comparison with [3]:** We also compared with PE-Net [3], which is a new architecture of SGG that utilizes prototype embedding methods for entities and their relationships. However, a direct comparison with PE-Net would be unfair as we used a very simple backbone model, e.g., Motif and VCTree, to apply the self-training framework. Hence, for fair comparisons, we implemented self-training to PE-Net and compared PE-Net+ST-SGG with PE-Net. Please note that we used vanilla PE-Net since PE-Net-Reweight was missing in the official code (https://github.com/VL-Group/PENET). Moreover, we trained PE-Net and PE-Net+ST-SGG with the same hyperparameters, and we fixed the hyperparameters for ST-SGG as: EMA $\alpha^{inc}=0.4$, EMA $\alpha^{dec}=0.4$, the loss coef. $\beta=1.0$, without searching them.
>
>
> |   **Task**  |                 | **PredCls**|              |                  |**SGCls**|                  |
> |:-------------:|:-------------:|:-------------|-------------:|:-------------:|:-------------:|:-------------:|
> | **Method** |  **R@50 / 100**  |  **mR@50 / 100** | **F@50 / 100** | **R@50 / 100**  |  **mR@50 / 100** | **F@50 / 100** |
> | PE-Net                             |    65.0 / 67.1  |  28.7 / 30.6   |   39.8 / 42.1   |    38.3 / 39.3    |   18.8 / 19.9  |    25.2 / 26.4  |
> |   **PE-Net + ST-SGG**    |  62.2 / 64.6    |  31.6 / 33.7   |   **41.9 / 44.3**   |    36.2 / 37.3    |    20.9 / 22.0 |    **26.5 / 27.7**  |
>
>
> In the above table, we observed that when ST-SGG is applied to PE-Net, the performance is increased in terms of F@K. This implies that the proposed ST-SGG appropriately assigns pseudo-labels to enhance the generalization of PE-Net. This result further demonstrates that our method is indeed model-agnostic to any SGG models.
>
> We have added [2] and [3] into our revised paper. Please see **BLUE** rows of Table 1 in **the modified PDF file.**
>
> Lastly, please understand that regarding the paper [1], as the proceeding of ICCV 2023 was released after the submission date of our paper, it was impossible to compare it with ours.
>
>
> [1] Vision Relation Transformer for Unbiased Scene Graph Generation, ICCV2023.
>
> [2] Stacked Hybrid-Attention and Group Collaborative Learning for Unbiased Scene Graph Generation, CVPR 2022.
>
> [3] Prototype-based Embedding Network for Scene Graph Generation, CVPR 2023.

---

### Official Review · Reviewer_NQzC · 2023-10-31

**Soundness:** 3 good
**Presentation:** 3 good
**Contribution:** 3 good
**Rating:** 8
**Confidence:** 5

**Summary:**

This paper proposes a novel self-training algorithm of training scene graph models based on un-annotated triplet loss. The distinction between un-annotated and background relationship class is drawn with the help of a class-specific dynamic threshold. Proposed method has been validated in Visual Genome dataset with two classic SGG baseline (MOTIF and VCTREE) and with four recent-most SGG baselines (Resam. IT-SGG, BGNN, and HetSGG). Proposed method increases the performance of the minority classes with slight decrease of majority classes.

**Strengths:**

This paper has the following strengths

**1. Well-motivated:** The paper's motivation is well-stated and clearly disseminated to the reader. Exploring vast un-annotated triplets in the image databases is one of the desired direction of open-set annotation dataset and Visual Genome is one of the most useful resources for conducting such exploration. This paper successfully established this motivation in their introduction.

**2. Novel self-training loss:** The proposed simple yet novel and effective loss is one of the major strengths of the paper. The discussion regarding challenges of fixed-threshold based distinction between background class and un-annotated class is interesting and highly relevant. The proposed class-specific thresholding solves the problem efficiently and demands further exploration.

**3. Less worsening of majority classes:** Traditional debiasing schemes of SGG usually hurt the recall classes significantly while improving the minority classes. However, this paper hurts the recall minimally, especially with the baseline MOTIF and VCTree.

**4. Detailed ablation study on thresholding:** Since the main contribution of this paper is a threshold-based self-training algorithm, authors have performed detailed analysis on how to choose appropriate threshold.

**Weaknesses:**

This paper has the following weaknesses

**1. Comparison with other debiasing methods missing.** The paper demonstrated that their method can improve the diversity of the baseline models. However, such improvement of baseline models is prevalent in SGG literature now. Therefore, a direct comparison with other debiasing methods would shed light more on their performances. For example,  VCTree+ST-SGG should be compared with VCTree+Resample. With that comparison, the readers would have better understanding of their methods performance.

**2. Zero-shot evaluation missing.** Since unannotated triplets are being accounted in the training, this method might improve the zero-shot performance of the baseline model as well. A separate study with zero-shot evaluation would strengthen the claim of the paper.

**Questions:**

The authors determine the threshold in an object-agnostic fashion considering only the relationship classes. However, the same relationship label has different semantic meaning to different pair and therefore an object-conditional viewpoint of relationship distribution may improve the result more.

To clarify more, in the loss calculation phase, if we consider the viability of a prediction conditioned on its object, we might have better choice on the threshold. For example, the prediction probability of 'on' will be higher for 'building-window' pair, however, it will be extremely low for 'man-pizza' pair. Is it possible to use such commonsense knowledge in the threshold choice of the loss equation?

**Post-rebuttal rating:** My concerns and queries have been properly addressed and therefore, I keep my original rating of 8.

---

> ### Author Response · Authors · 2023-11-16
>
> **(W1) Lack of comparison with debiasing methods. e.g., VCTree + Resamp**
>
> Please note that as for the specific example that the reviewer mentioned (i.e., comparison between VCTree+ST-SGG and VCTree+Resample [1]), we have indeed compared them in rows 18 and 19 of Table 1.
>
> Upon the reviewer’s request, we provide analysis on the comparisons with debiasing methods presented in Table 1. We observe that when ST-SGG is applied to Motif/VCTree, it shows a lower F@K compared with the case when Resampling is applied to Motif/VCTree. This is primarily attributed to the severe long-tailed distribution in the SGG nature, and as a result, a straightforward resampling approach appears to be more effective in addressing the imbalance of relation classes. However, we observe that applying both Resaming and ST-SGG to Motif/VCTree significantly outperforms Motif/VCTree+Resam. This implies that utilizing the pseudo-label can further enhance the model’s generalization even when resampling is applied.
>
> Based on this result, we argue that ST-SGG stands out as an effective method for improving the generalization capabilities of SGG models. Its ease of implementation and compatibility with other debiasing techniques are notable advantages. Specifically, ST-SGG requires only the addition of missing annotations and the computation of an extra loss component associated with pseudo-labeled predicates. These attributes underscore ST-SGG’s potential as a versatile and potent tool in the field of SGG.
>
>
> [1] Bipartite Graph Network with Adaptive Message Passing for Unbiased Scene Graph Generation. CVPR’21. Ronjie Li et al.
>
>
> **(W2) Zero-shot performance of the baseline model as well**
>
> We thank the reviewer for giving insightful feedback, which has illuminated aspects of our work that were previously unaddressed in the paper. We conducted an additional experiment to evaluate the zero-shot performance. Following the zero-shot settings [1],  we compute the recall@K values for specific <subject, predicate, object> pairs, which are not included in the training set but emerge in the testing phase. This experiment is conducted under the PredCls task.
>
>
> | Method | zR@50 / 100 | Method | zR@50 / 100 |
> |----------|----------|----------|----------|
> | Motif | 16.3 / 19.4 | VCTree | 16.7 / 19.7 |
> | Motif + Resampling | 17.0 / 19.5 | VCTree + Resampling | 16.3 / 19.4 |
> | Motif + IE-Trans | 13.0 / 15.9 | Motif + IE-Trans | 14.0 / 16.5 |
> | **Motif + ST-SGG** | **18.0 / 21.0** | **Motif + ST-SGG** | **19.8 / 21.0** |
>
>
>
> The above table reveals that debiasing techniques, including resampling and IE-Trans, do not enhance the zero-shot performance of SGG models. To elaborate, the limitation of the resampling approach lies in its simplistic alteration of the number of instances involved in training for each predicate class, which does not augment any variance in the data. As for IE-Trans, it generates pseudo-labels only for tail predicates classes, which restricts the ability of finding zero-shot triplets. On the other hand, the application of ST-SGG shows improvements in the zero-shot performance for both Motif and VCTree models. This improvement suggests that ST-SGG can generate pseudo-labels that are instrumental in enhancing the model's generalization to unseen triplet patterns. The effectiveness of ST-SGG in this context underscores its potential as a robust tool for improving the generalization capabilities of SGG models, where the ability to predict novel patterns is crucial.
>
> We would like to extend our appreciation to the reviewer once again for suggesting the point that strengthens the advantage of our work. We have updated these results in Appendix E.5 of **the modified PDF file.**
>
>
> [1] Unbiased Scene Graph Generation from Biased Training. Tang et al. CVPR’20.

---

> ### Author Response · Authors · 2023-11-16
>
> **(Q1) Object-conditional relation class thresholding. (based on commonsense knowledge)**
>
> We agree with the reviewer’s suggestion that object-conditional relation pseudo-labeling is also a promising direction of self-training for SGG. The object-conditional pseudo-labels are expected to strengthen our approach by reflecting commonsense knowledge to decide whether pseudo-labeling or not. We suggest two approaches based on the SGG works with commonsense knowledge [1, 2].
>
> - One approach would be to utilize the frequency of triplets as a commonsense knowledge, which is obtained from counting the appearance of <subject, predicate, object> in the training dataset [1]. For example, if a predicate “on” appears more frequently between “man-horse” than between “man-pizza” in the training data, then we encourage the pseudo-label <man, on, horse> to contribute more to the loss than the pseudo-label <man, on, pizza>, given that these two pseudo-labels are obtained by ST-SGG. However, although this simple approach can enhance the accuracy of pseudo-label generation, it is not expected to improve the zero-shot performance as it allows pseudo labels to be assigned only when the generated triplets emerge in the training set.
> - Another approach would be to utilize the knowledge graph from external sources. By comparing the triplet in the commonsense knowledge graph, such as ConceptNet [3], we can confirm the generated pseudo-labels are valid or not. In this case, we can reduce the number of incorrect pseudo-labels by filtering the pseudo-label based on commonsense.
>
> We express our gratitude to the reviewer for bringing to light issues that we had overlooked, thanks to their valuable insights. We acknowledge the significance of these issues and will consider them as part of our future work.
>
> [1] Neural Motifs: Scene Graph Parsing with Global Context. Zellers et al. CVPR’18.
>
> [2] Visual Distant Supervision for Scene Graph Generation. Yuan Yao et al. ICCV’21.
>
> [3] Conceptnet 5.5: An open multilingual graph of general knowledge. Robyn Speer et al. AAAI-17

---

> > ### Comment · Reviewer_NQzC · 2023-11-23
> > **Final Rating**
> >
> > All of my concerns have been addressed and therefore I retain my original rating of 8.

---

### Official Review · Reviewer_jErj · 2023-11-03

**Soundness:** 3 good
**Presentation:** 3 good
**Contribution:** 3 good
**Rating:** 6
**Confidence:** 3

**Summary:**

The paper proposed the ST-SGG framework to address the long-tailed predicate issue in Scene Graph Generation (SGG). It incorporates the CATM pseudo-labeling method and a Graph Structure Learner (GSL). Experimental results confirm improved performance on fine-grained predicate classes.

**Strengths:**

- ST-SGG serves as a model-agnostic framework, meaning it can be applied to various existing SGG models. This aspect has the potential to expand the applicability of the self-training in SGG.
- Based on experimental results, the proposed framework seems to effectively alleviate the issues of long-tailed distribution. The performance improvements are primarily concentrated on fine-grained predicate classes.

**Weaknesses:**

- In some experiments, the R@k values significantly decreased after employing the proposed framework. The paper lacks some explanations for this phenomenon.
- There's a concern if the proposed framework might, in some scenarios, sacrifice a considerable amount of overall performance to achieve improvement in fine-grained predicates.
- The paper lacks a clear depiction of the overall framework structure. Please clearly demonstrate how the different components interact and how pseudo-labels are generated and applied.

**Questions:**

See the Weaknesses.

---

> ### Author Response · Authors · 2023-11-16
>
> **(W1) Explanation of the decreased R@K when ST-SGG is applied.**
>
> The decrease in R@K is associated with the trade-off between R@K and mR@K. In other words, if a model deliberately lowers the number of predictions for head predicate classes (e.g., 'on') while increasing it for tail predicate classes (e.g., 'standing on', 'walking on', and 'walking in'), we would encounter a decrease in R@K and an increase in mR@K. That being said, we can deliberately increase R@K at the expense of reduced mR@K and vice versa.
>
> Therefore, to mitigate this issue, we also reported the F@K [1, 2] that is designed to evaluate models by considering the balance between R@K and mR@K (i.e., harmonic mean of R@K and mR@K). We would like to emphasize that **applying ST-SGG consistently outperforms baselines in terms of F@K,** demonstrating the superiority of ST-SGG regardless of the trade-off between R@K and mR@K. Moreover, it is important to note that the F@K values obtained from the validation set are used for model selection including parameter and hyperparameter determination, and that we reported R@K and mR@K values when the validation F@K is the best.
>
> We thank the reviewer for giving the valuable feedback, which enhances the quality of our paper. To clarify the explanation of our results, we elaborated on the description of F@K in Appendix E.2 of **the modified PDF** file with **BLUE** text.
>
> [1] Qianyu Chen et al. Fine-grained scene graph generation with data transfer. In ECCV, 2022.
>
> [2] Khandelwal, S. et al (2022). Iterative scene graph generation. Advances in Neural Information Processing Systems.
>
>
>
> **(W2) Concern of sacrificing a considerable amount of overall performance.**
>
> As we have mentioned in our response to W1, we could deliberately increase R@K at the expense of a decrease in mR@K, if we are concerned about too low R@K. However, since doing so would make it hard to correctly evaluate the performance of our model, we instead relied on F@K. Once again, we reported R@K and mR@K when the validation F@K is the best.
>
> Nevertheless, we would like to draw attention again to the experimental result on the OI-V6 dataset. In Table 8 of **the modified PDF**, applying ST-SGG to SGG models leads to improvement in both R@K and mR@K. This indicates that our proposed approach is effective in enhancing model’s generalization across both head and tail predicate classes, implying that our approach does not compromise the overall performance. Instead, we argue that ST-SGG fulfills the primary objective of self-training: boosting the model’s generalization ability through the utilization of unlabeled instances. This achievement underscores the robustness and effectiveness of our proposed methodology.
>
> **(W3) Depiction of the overall framework structure**
>
> We appreciate the reviewer’s suggestion to enhance the presentation quality of our paper.  In response, we have revised the description of ST-SGG with Algorithm 1 of Appendix C.1 in **the modified PDF** with **BLUE** text.

---

### Meta-Review · Area_Chair_MPGu · 2023-12-10

**Metareview:**

Paper proposes a scene graph generation approach which leverages self-training to enhance performance on, particularly, long tail predicates. Approach uses class-specific pseudo-labeling and is generic -- able to work with any underlying scene graph generation model. It is shown to be useful and improve performance when combined with a number of scene graph generation models.

Paper was reviewed by three reviewers that gave it the following ratings: 1 x Marginally above the acceptance threshold, 1 x Accept, good paper, and 1 x Marginally below the acceptance threshold. Reviewers agree that the approach is well motivated, intuitive, the paper is well written and contains ample ablations. Main concerns were with respect to lower Recall, lack of direct comparisons to re-sampling baselines, lack of illustration for overall framework and overall novelty. Authors have addressed majority of these concerns to the satisfaction of reviewers.

AC has read the reviews, rebuttal, discussion that followed and the paper itself. AC agrees with most comments raised by reviewers and the fact that authors have addressed them appropriately in the rebuttal and through revisions. While approach is indeed a form of adaptation of self-training that perviously appeared in other domains, e.g., image classification, adopting it to scene graph generation task is highly not-trivial. Authors do a good job at highlighting the issues that arise and solving them throughout the paper. Hence, in AC's opinion, the lack of novelty isn't a significant downside of proposed work and it would be a valuable contribution to ICLR.

**Justification For Why Not Higher Score:**

As mentioned by the reviewers the approach is an adaptation of self-training to a specific task -- scene graph generation. It is unclear whether or how it could be translated to other domains (e.g., object detection, segmentation).

**Justification For Why Not Lower Score:**

While the novelty of proposed approach with pseudo-labeling and self-training isn't earth shattering (i.e., is adopted from image classification), the application of such techniques to scene graph generation is by no means trivial. The paper does a  good job at both discussing and addressing impending issues and the approach has clear merit.

---

### Decision · Program_Chairs · 2024-01-16

Accept (poster)